# Quantum advantage and stability to errors in analogue quantum simulators

Rahul Trivedi [1,2,3] ✉, Adrian Franco Rubio [1,2] ✉ & J. Ignacio Cirac [1,2] ✉

Several quantum hardware platforms, while being unable to perform fully fault-tolerant quantum computation, can still be operated as analogue quantum simulators for addressing many-body problems. However, due to the presence of errors, it is not clear to what extent those devices can provide us with an advantage with respect to classical computers. In this work, we make progress on this problem for noisy analogue quantum simulators computing physically relevant properties of many-body systems both in equilibrium and undergoing dynamics. We first formulate a system-size independent notion of stability against extensive errors, which we prove for Gaussian fermion models, as well as for a restricted class of spin systems. Remarkably, for the Gaussian fermion models, our analysis shows the stability of critical models which have long-range correlations. Furthermore, we analyze how this stability may lead to a quantum advantage, for the problem of computing the thermodynamic limit of many-body models, in the presence of a constant error rate and without any explicit error correction.

Quantum information processing systems hold the promise of solving a number of problems in physics and computer science faster than their classical counterparts[1,2]. However, most quantum algorithms with theoretical performance guarantees require a fault-tolerant quantum computer[3–5]. While in principle possible, implementing a fault tolerant quantum computer is a technological challenge that could still take a long time to solve. This has motivated several investigations trying to identify both quantum algorithms, as well as physically relevant computational problems, that can be addressed by quantum hardware in the near term and without any explicit error correction.

Analog quantum simulators, wherein a target Hamiltonian is mimicked by an experimentally controllable system, have shown some promise in solving problems arising in many-body physics in the near term[6–8]. A typical analog quantum simulator, while not necessarily being able to perform an arbitrary computation, would instead aim to approximately implement a relevant spatially-local Hamiltonian, $H$. In several many-body problems, $H$ can additionally be taken to be translationally invariant. The quantum simulator can then be used to prepare a physically

relevant quantum state $\rho_H$ associated with the Hamiltonian $H$, such as its ground state, Gibbs state or a state produced under dynamics. In a typical quantum simulation experiment, we would then measure the expectation value of an intensive observable $O$, which is either often a local observable at a single site on the lattice or a correlation function, i.e., a product of local observables at a few sites. Examples of such Hamiltonians and observables can be found in a variety of problems in physics—for e.g., in study of correlated electronic systems[9–12], quantum spin systems[13–17] as well as lattice-gauge theories[18,19]. From a more experimental standpoint, there have been several proposals to implement quantum simulation of these models in different hardware platforms, such as cold atoms in optical lattices, trapped ion systems or superconducting qubits[20–28].

Practically, quantum simulators offer several distinct advantages in solving many-body problems as opposed to general purpose quantum computers. First, quantum simulators aim to solve only a much smaller and specialized set of problems, and thus have much milder hardware requirements than a universal quantum computer. Furthermore, quantum simulators are more

[1]Max-Planck-Institut für Quantenoptik, Garching, Germany. [2]Munich Center for Quantum Science and Technology (MCQST), Munich, Germany. [3]Electrical and Computer Engineering, University of Washington, Seattle, WA, USA. ✉e-mail: rahul.trivedi@mpq.mpg.de; adrian.franco@mpq.mpg.de; ignacio.cirac@mpq.mpg.de

naturally suited to many-body problems, since they avoid a Hamiltonian-to-circuit mapping e.g., by trotterizing the evolution into a quantum circuit, which typically incurs in a rapid proliferation of errors[7,8,29–31]. Additionally, since the observables of interest are typically local intensive observables, we expect them to be somewhat more robust to errors even if the global quantum state of the simulator is very sensitive. These expectations make quantum simulators very promising in providing some advantage with respect to classical computers when addressing typical quantum many-body physics problems.

However, developing rigorous criteria to outline the quantum advantage of a quantum simulator runs into several theoretical issues. First, quantum simulators do not implement any error correction and typically simulate many-body physics in the presence of noise. While several previous works have theoretically outlined the computational power of noiseless quantum simulators by developing the notion of a universal quantum simulator[32–34] and rigorously established the possibility of quantum advantage, the presence of experimentally realistic noise has to be carefully accounted for in understanding their utility in many-body problems. Second, since quantum simulators are usually devoted to analyzing intensive observables, and in many-body physics we are typically interested in the thermodynamic limit of such observables, we need to revisit the usual notion of quantum advantage. In particular, instead of characterizing the quantum and classical effort required to compute the many-body observable as a function of the system size, which is not meaningful in the thermodynamic limit, we can characterize the effort required to compute the many-body intensive observable within a user-specific precision of the thermodynamic limit[35,36].

In this work, we address both of these issues—we provide evidence that many physically relevant critical and non-critical many-body models are stable to errors in the quantum simulator. Importantly, even without error correction, we can use a quantum simulator to determine the thermodynamic limits of intensive observables in these problems to a hardware-limited precision. Furthermore, we also propose a notion of quantum advantage, in the presence of errors, for such problems, where the figure of merit is the computational time to obtain an intensive quantity in the thermodynamic limit to a hardware-limited precision. By providing explicit lower bounds on certifiable classical algorithms for the many-body problems that we consider, we provide evidence that quantum simulators can possibly provide superpolynomial to exponential quantum advantage over rigorous classical algorithms even without error correction.

## Results

### Setup

To keep our analysis general, we will consider quantum simulators for solving both closed system (i.e., implementing a Hamiltonian) as well as open system (i.e., implementing a Lindbladian) many-body problems. Suppose that the quantum simulator was trying to configure a spatially local Lindbladian $\mathcal{L}$ on $n$ spins given by

$$\mathcal{L} = \sum_\alpha \mathcal{L}_\alpha, \tag{1}$$

where $\mathcal{L}_\alpha$ is a Lindbladian acting on spins within a local region $\Lambda_\alpha$. For translationally invariant problems, the superoperator $\mathcal{L}_\alpha$ would additionally be independent of $\Lambda_\alpha$, and for closed system problems, we can assume $\mathcal{L}_\alpha = -i[h_\alpha, \cdot]$ for some operator $h_\alpha$ supported on $\Lambda_\alpha$. The quantum simulator would, in general, suffer from coherent errors in the configured Lindbladian as well as incoherent errors arising due to its interaction with an external environment. As depicted in Fig. 1, to account for these errors, we model the 'implemented' Lindbladian on the quantum simulator by

$$\mathcal{L}'(t) = \sum_\Lambda \left( \mathcal{L}'_\alpha - i[h_{SE,\alpha}(t), \cdot], \right). \tag{2a}$$

Here $\mathcal{L}'_\alpha - \mathcal{L}_\alpha$ is an error in the Lindbladian implemented on spins in $\Lambda_\alpha$—this can arise either from configuration errors in the Hamiltonian and the jump operators corresponding to $\mathcal{L}_\alpha$, or from incoherent errors that can be well approximated as Markovian. Furthermore, we also consider the possibility of non-Markovian incoherent errors—these are captured by $h_{SE,\alpha}(t)$, which accounts for the interaction of the spins in the region $\Lambda_\alpha$ with an external environment (in the interaction picture with respect to the environment). For concreteness, we will model the decohering environment as a Gaussian environment and assume that

$$h_{SE,\alpha}(t) = \sum_{j=1}^{n_L} A_{j,\alpha}(t) Q_{j,\alpha}^\dagger + \text{h.c.}, \tag{2b}$$

where $Q_{1,\alpha}, Q_{2,\alpha} \ldots Q_{n_L,\alpha}$ are the jump operators, each supported on $\Lambda_\alpha$, through which the spins in $\Lambda_\alpha$ interact with a decohering environment, and $A_{1,\alpha}(t), A_{2,\alpha}(t) \ldots A_{n_L,\alpha}(t)$ are annihilation operators for the environment. We assume that $[A_{j,\alpha}(t), A^\dagger_{j',\alpha'}(t')] = \delta_{\alpha,\alpha'} \delta_{j,j'} K_{j,\alpha}(t - t')$ for bosonic environments or $\{A_{j,\alpha}(t), A^\dagger_{j',\alpha'}(t')\} = \delta_{\alpha,\alpha'} \delta_{j,j'} K_{j,\alpha}(t - t')$ for

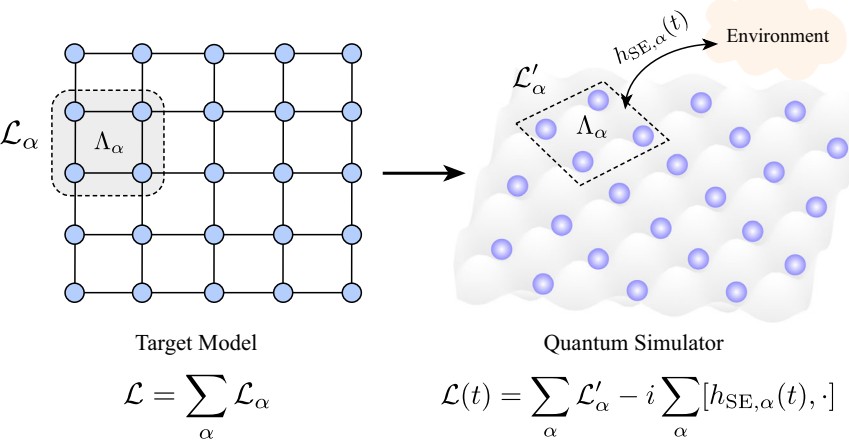

Target Model

$$\mathcal{L} = \sum_\alpha \mathcal{L}_\alpha$$

Quantum Simulator

$$\mathcal{L}(t) = \sum_\alpha \mathcal{L}'_\alpha - i \sum_\alpha [h_{SE,\alpha}(t), \cdot]$$

**Fig. 1 | Schematic depiction of our error model for analog quantum simulator.** A target Lindbladian $\mathcal{L}$, expressed as sum of Lindbladian terms modeling interactions between groups of spins, when implemented on an analog quantum simulator would have an hardware error per qubit—this error can either be due to an incorrect configuration of the Lindbladian (i.e., implementing $\mathcal{L}'_\alpha$ instead of $\mathcal{L}_\alpha$) or due to interaction with an external decohering environment.

fermionic environments and we choose the normalization of $Q_{j,\alpha}$ such that $\int_{\mathbb{R}}|K_{j,\alpha}(\tau)|d\tau = 1$. The function $K_{j,\alpha}(\tau)$ can be understood as the memory kernel corresponding to the non-Markovian system-environment interaction. In particular and not unexpectedly, choosing $K_{j,\alpha}(\tau) = \delta(\tau)$ would yield a Markovian master equation for an environment initially in the vacuum state.

We introduce a parameter $\delta$ such that $\|\mathcal{L}'_\alpha - \mathcal{L}_\alpha\|_\diamond \leq \delta$ (recall that the diamond norm $\|\mathcal{S}\|_\diamond$ of a superoperator $\mathcal{S}$ is the smallest number $s$ that satisfies $\|\mathcal{S}(\rho)\|_1 \leq s$ for all $\rho$ which are possibly defined on a system larger than the support of $\mathcal{S}$) and $\|Q_{j,\alpha}\| \leq \sqrt{\delta}$—the parameter $\delta$ can be considered to be the "hardware error rate" in the quantum simulator. A well designed experimental setup can, in principle, achieve $\delta \ll 1$—however, since there are an extensive number of errors in the simulator, we generically expect the state of the simulator to be at a distance of $\delta \times n$ from the target state. In the worst case, this would imply that the results of the quantum simulator can only be trusted when $\delta < o(1/n)$, and this would limit their applicability to small-scale problems. Importantly, for applications of quantum simulators to problems in many-body physics, this would imply that noisy quantum simulators, in the worst case, cannot be used to faithfully capture thermodynamic limits (i.e., $n \to \infty$).

An alternative viewpoint would be to ask if there are certain interesting many-body problems for which a good estimate for the thermodynamic limit can be produced with a hardware with constant errors. This motivates us to look for 'quantum simulation tasks' which are stable to these extensive errors, as made precise in the following definition.

**Definition 1.** (Stable quantum simulation task). Consider the quantum simulation task on $n$ spins of measuring an observable $O$ with $\|O\| \leq 1$ in a state $\rho_\mathcal{L}$ associated with a target Lindbladian $\mathcal{L}$. This task is said to be stable if the corresponding state $\rho'_\mathcal{L}$ prepared by the noisy simulator satisfies

$$|\mathrm{Tr}(O\rho_\mathcal{L}) - \mathrm{Tr}(O\rho'_\mathcal{L})| \leq f(\delta),$$

for some continuous $f$ of the hardware-error rate $\delta$, independent of $n$, such that $f(\delta) \to 0$ as $\delta \to 0$.

If a quantum simulation task is stable as per this definition, we can hope to be able to estimate the thermodynamic limit of the observable on a quantum simulator to a precision limited only by the hardware error rate $\delta$, and independent of the size of the problem. In particular, these problems would not require the hardware error to be scaled down with system size even in the absence of error correction, and it is reasonable to consider them to be problems that analog quantum simulators can conceivably solve in the near term.

In the remainder of this section, we systematically study several important problems arising in many-body physics, and show that commonly considered intensive observables are expected to be stable to errors. We first study geometrically local Gaussian fermion models with Gaussian errors—these models, while being a restricted class of many-body models, provide a setting where we are able establish strong stability results. Furthermore, physically we expect the results of these models to provide evidence of stability in more general situations. For these models, we show that intensive observables (either local observables, or translationally invariant sums of local observables) are stable both for the problem of constant-time dynamics and equilibrium without any restrictive assumptions on the model—our results hold not only for gapped models, but also for gapless models. Then, we examine the same question for (non-Gaussian) many-body spin systems—here, we rely on, and in some cases extend, stability results that show local observables are stable in constant-time dynamics and in equilibrium, but with more restrictive

**Table 1 | Summary of the stability results for dynamical and equilibrium many-body problems, together with the required assumptions on the many-body model and errors**

| Problem | Error | Assumption | Observable | Stability, $f(\delta)$ |
|---|---|---|---|---|
| Dynamics | General errors | GF: None. | GF: $k$-local observables. | GF: $O(\delta t)$. |
| | | SS: None. | SS: $k$ – local observables. | SS: $O(\delta t^{e^{at}})$. (Established in ref. 38 for Markovian errors). |
| Ground state | Coherent Hamiltonian errors | GF: Assumption 1. | GF: Translationally invariant $k$ – local observables. | GF: $O(\delta^\beta)$, where $\beta$ is a model dependent constant. (Established in ref. 37 for gapped models). |
| | | SS: Stable gap. | SS: Local observables | SS: $O(\delta)$ (Established in ref. 54). |
| Gibbs state | Coherent Hamiltonian errors | GF: No assumption. | GF: Translationally invariant $k$ – local observables. | GF: $O(\sqrt{\delta})$. |
| | | SS: Stably exponentially clustered correlations. | SS: Local observables. | SS: $O(\delta \log^d(1/\delta))$. (Established in ref. 46). |
| Fixed points | GF: Coherent and Incoherent Markovian errors. | GF: Assumption 2. | GF: Translationally invariant $k$ – local observables | GF: $O(\delta^\beta)$, where $\beta$ is a model dependent constant. |
| | SS: General errors. | SS: Rapid Mixing. | SS: Local observables. | SS: $O(\delta)$. (Established in ref. 38 for Markovian errors). |

Both results for Gaussian fermions and spin systems are summarized—"GF" indicates Gaussian fermions and "SS" indicates spin systems. Note that observables for the Gaussian fermionic problems are all quadratic.

assumptions on the system. Our results are summarized in Table 1, and lend strong evidence for several many-body problems being amenable to noisy quantum simulation.

## Stability of Gaussian fermion models

We will consider fermions arranged on a $d$−dimensional lattice with $L$ sites in each direction $\mathbb{Z}_L^d$, and at each site we have $D$ fermionic modes −we denote by $c_x^\alpha$ for $x \in \mathbb{Z}_L^d$, $\alpha \in \{1, 2 \ldots 2D\}$ the Majorana operators associated with each site $x$. We consider a general open quantum simulation problem with geometrically local interactions with interaction range $R$. This is specified by a quadratic Hamiltonian $H$, and $n_L$ linear jump operators $L_{j,x}$ for every site $x \in \mathbb{Z}_L^d$,

$$H = \sum_{\substack{x,y \in \mathbb{Z}_L^d \\ d(x,y) \le R}} \sum_{\alpha,\beta=1}^{2D} h_{x,y}^{\alpha,\beta} c_x^\alpha c_y^\beta, \tag{3a}$$

$$L_{j,x} = \sum_{\substack{y \in \mathbb{Z}_L^d \\ d(x,y) \le R}} \sum_{\alpha=1}^{2D} l_{j;x,y}^\alpha c_y^\alpha, \forall j \in \{1, 2 \ldots n_L\}. \tag{3b}$$

Without loss of generality, we can assume that $|h_{x,y}^{\alpha,\beta}| \le 1$, $|l_{j;x,y}^\alpha| \le 1$ and $n_L \le 2D(2R+1)^d$.

For the results in this subsection, we restrict ourselves to Gaussian errors (coherent or incoherent) when this model is implemented on a quantum simulator. Due to coherent hardware errors, the quantum simulator instead implements a perturbed Gaussian fermion Hamiltonian $H'$,

$$H' = \sum_{\substack{x,y \in \mathbb{Z}_L^d \\ d(x,y) \le R}} \sum_{\alpha,\beta=1}^{2D} h_{x,y}'^{\alpha,\beta} c_x^\alpha c_y^\beta, \tag{4a}$$

such that $|h_{x,y}^{\alpha,\beta} - h_{x,y}'^{\alpha,\beta}| \le \delta$. Furthermore, due to errors in the configuration of the jump operators, or due to Markovian incoherent errors, the quantum simulator implements perturbed jump operators $L'_{j,x}$,

$$L'_{j,x} = \sum_{\substack{y \in \mathbb{Z}_L^d \\ d(x,y) \le R}} \sum_{\alpha=1}^{2D} l_{j;x,y}'^\alpha c_y^\alpha, \forall j \in \{1, 2 \ldots n_L\}. \tag{4b}$$

where again $|l_{j;x,y}^\alpha - l_{j;x,y}'^\alpha| \le \delta$. Furthermore, we also consider Gaussian incoherent interactions with a decohering environment which, following the general setup described previously, is captured by a Gaussian system-environment $H_{SE}(t) = \sum_{x \in \mathbb{Z}_L^d} h_{x,SE}(t)$ with

$$h_{x,SE}(t) = \sum_{j=1}^{n_L} A_{j,x}(t) Q_{j,x}^\dagger + \text{h.c.}. \tag{5a}$$

Here

$$Q_{j,x} = \sum_{\substack{y \in \mathbb{Z}_L^d \\ d(x,y) \le R}} q_{j,x;y}^\alpha c_y^\alpha, \tag{5b}$$

with $|q_{j,x;y}^\alpha| \le \sqrt{\delta}$ and $A_x(t)$ is an annihilation operator in the fermionic environment coupling to sites in the neighborhood of $x$. These annihilation operators satisfy $\{A_x(t), A_{x'}^\dagger(s)\} = \delta_{x,x'} K_x(t-s)$, where $K_x(\tau)$ is the memory kernel describing the system-environment interaction and is assumed to satisfy $\int_{\mathbb{R}} |K_x(\tau)| d\tau \le 1$.

**Finite-time dynamics.** We first consider the problem of evolving the quantum simulator for time $t$ and measure the expectation value of Gaussian observables $O_0$ which are either $k$−local, i.e., they act on a set $\mathcal{S} \subseteq \mathbb{Z}_L^d$ of $k$ sites

$$O_0 = \sum_{x,y \in \mathcal{S}} \sum_{\alpha,\beta=1}^{2D} o_{x,y}^{\alpha,\beta} c_x^\alpha c_y^\beta, \tag{6}$$

or weighted averages of $k$−local Gaussian observables i.e., are of the form $\sum_{i=1}^M w_i O_i$, where $O_i$ is of the form Eq. (6), $\sum_{i=1}^M |w_i| = 1$ and $M$ can possibly grow with $n = DL^d$. We consider an arbitrary Gaussian initial state, and let the target state $\rho$ be the state obtained by evolving it with the target Lindbladian specified by Eq. 3 for time $t$. We show the following proposition in supplemental note IIA.

**Proposition 1.** The quantum simulation task of measuring $k$−local Gaussian observables, or their weighted sums, after constant-time dynamics under a spatially local Gaussian Hamiltonian is stable to coherent and incoherent Gaussian errors modeled by Eqs. 4 and 5 with $f(\delta) = O(t\delta)$.

We point out that the dependence of the error between the observable in perturbed and unperturbed models on $t$ is independent of the dimensionality of the lattice $d$−this result is thus stronger than what would be expected simply from locality, wherein the error would be expected to grow as $t \times$ (Number of sites in the light cone at time $t$) $\propto t^{d+1}$−we revisit this in "Methods".

**Equilibrium.** We next study the stability properties of intensive observables in equilibrium. The observables for which we establish stability results for equilibrium properties are more restrictive then the observables studied in the problem of dynamics−in particular, we assume that the observables are translationally invariant Gaussian observables generated by averaging the observables in Eq. (6). More specifically, if $O_0$ is a $k$−local Gaussian observable of the form of Eq. (6), then we consider observables of the form

$$O = \frac{1}{n} \sum_{x \in \mathbb{Z}_L^d} \tau_x(O_0), \tag{7}$$

where $\tau_x(O_0)$ is the observable $O_0$ translated by $x$. Physically, such observables correspond to intensive observables which are obtained on averaging contributions different points on the lattice e.g., the average number of fermions $\sum_{x \in \mathbb{Z}_L^d} a_x^\dagger a_x / L^d$. Furthermore, often in many-body physics, the equilibrium state of interest $\rho$ is itself translationally invariant i.e., $\tau_x(\rho) = \rho$ for all $x \in \mathbb{Z}_L^d$. In this case, the expected value of a $k$−local observable $O_0$ coincides with the expected value of the local observable $O$ generated by $O_0$ as per Eq. (7) since

$$\text{Tr}(\rho O) = \frac{1}{n} \sum_{x \in \mathbb{Z}_L^d} \text{Tr}(\rho(0)\tau_x(O_0)),$$
$$= \frac{1}{n} \sum_{x \in \mathbb{Z}_L^d} \text{Tr}(\tau_{-x}(\rho)(O_0)) = \text{Tr}(\rho O_0).$$

Consequently, for translationally invariant Hamiltonians, observables of the form in Eq. (7) are not more restrictive than those considered in dynamics (Eq. (6)).

For simplicity, we first consider the closed-system setting and study the stability of the ground state and Gibbs state. Suppose that the quantum simulator implements a target geometrically local Hamiltonian (Eq. 3a), but due to the presence of coherent errors instead configures a perturbed Hamiltonian $H'$ (Eq. 4a). We emphasize that we only consider the simpler coherent Hamiltonian errors while studying the stability of ground state and Gibbs state, since these

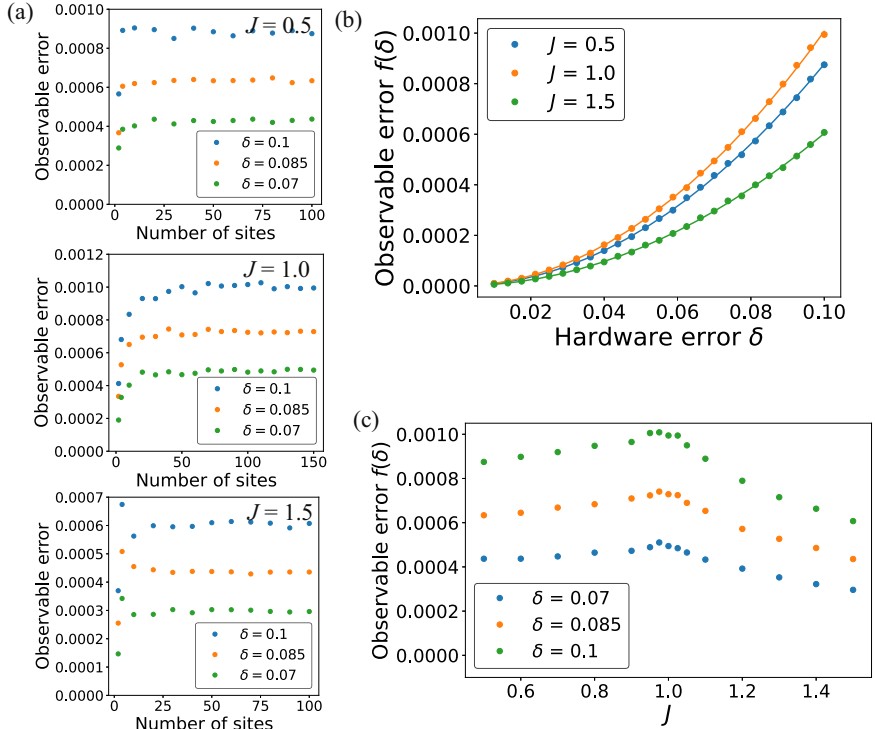

**Fig. 2 | Numerical study of the impact of errors in the SSH model.** The observable that we study here is $O = H_{SSH}[J]/n$, where $H_{SSH}[J]$ is the Hamiltonian of the ideal SSH model (Eq. (9)) **a** The error in the expected value of the observable $O$ in the ground state between the perturbed and unperturbed Hamiltonians, as a function of $\delta$, the hardware error, and the number of sites $n$. For both gapped ($J = 0.5, 1.5$) and gapless ($J = 1.0$) cases, we see that the error in $O$ becomes independent of $n$ as $N \to \infty$. **b** Numerically extracted error between the perturbed and unperturbed models for $n \to \infty$ as a function of $\delta$, and its fit with $\delta^2$. **c** The error between the perturbed and unperturbed model as a function of $J$—for the same hardware error $\delta$, this error peaks at $J = 1$ which is also the point at which the gap in the unperturbed model closes. All the errors are computed by averaging over 500 random instances of perturbed models.

states can only be meaningfully defined for a closed system. Later in this section, we will study the more natural equilibrium problem of the 'Lindbladian fixed point' where incoherent errors can also be accounted for. Furthermore, we make the following additional physically reasonable assumption on the density of modes of the target Hamiltonian $H$.

**Assumption 1.** The number of eigenfrequencies $n_\eta$ of $H$, which are eigenvalues of the matrix $h_{x,y}^{\alpha,\beta}$ defining the target Hamiltonian, lying in the interval $[-\eta, \eta]$ for sufficiently small $\eta$ satisfy the upper bound

$$n_\eta \leq nf_h(\eta) + \kappa(\eta, n),\tag{8}$$

where $n = DL^d$ is the number of fermionic modes, $f_h(\eta) \leq O(\eta^\alpha)$ as $\eta \to 0$ for some $\alpha > 0$ and $\kappa(\eta, n)$ is $o(n)$ for any fixed $\eta$.

Alternatively stated, this assumption is a continuity condition on the thermodynamic limit of the fraction of eigenmodes with energies in the interval $[-\eta, \eta]$ and it ensures that eigenvalues do not accumulate too fast near zero—more precisely, it demands that $\lim_{n \to \infty} n_\eta/n$ is a Hölder continuous function of $\eta$. It is expected to be true for most physically relevant models—in particular, it is weaker than the existence of a gap and thus contains gapped models, which are well known to exist for many experimentally relevant many-body problems. For a gapped Gaussian fermionic model, $f_h(\eta) = 0$ since, if there are fermionic eigenmodes near 0, then adding a fermion into these modes would provide an excited state with only $O(n^{-1})$ energy higher than the ground state energy. Furthermore, we also expect this assumption to be generically true for translationally invariant local Hamiltonians, where the eigenfrequencies can be described by a smooth dispersion relation $\omega(k)$ as a function of the momentum $k$ associated with that

mode. In this case, $f_h(\eta) \leq O(\eta^\alpha)$, with $\alpha$ being determined by the derivatives of $\omega(k)$ in the vicinity of $\omega = 0$.

Considering the target state to be the ground state of the Gaussian fermion Hamiltonian satisfying assumption 1, we then obtain the following proposition (proved in supplemental note IIB).

**Proposition 2.** The quantum simulation task of measuring translationally invariant Gaussian observables generated $k$−locally as described by Eq. (7), in the ground state of a spatially local Gaussian fermion Hamiltonian whose distribution of modes satisfies Eq. (8) is stable to coherent Hamiltonian errors as modeled by Eq. 4a with $f(\delta) = O(\sqrt{\delta}) + f_h(O(\delta^{1/4})) \leq O(\delta^\beta)$ for some model-dependent constant $\beta$.

We re-emphasize that the stability result above holds with only a mild continuity assumption on the density of modes of the model. In addition to gapped Gaussian fermion models whose stability has been previously shown[37], it holds for models which are not gapped, i.e., the energy separation between the ground state and the first excited state vanishes as $n \to \infty$. As an example, consider the ground state of the 1D Su–Schrieffer–Heeger (SSH) model on $n$ fermions with periodic boundary condition:

$$H_{SSH}[J] \equiv \sum_{i=1}^{n} t_i a_i^\dagger a_{i+1} + \text{H.c}, \quad t_i = \begin{cases} 1 & i \text{ odd}, \\ J & i \text{ even}. \end{cases}\tag{9}$$

where $a_{n+1} \equiv a_n$. This model displays a (topological) phase transition at $J = 1$, where the gap closes as $1/n$, and is gapped otherwise. The observable we consider is the energy density $H_{SSH}[J]/n$ of the unperturbed Hamiltonian. Figure 2(a) shows impact of changing system size on this energy density—we see that for both gapped ($J = 0.5, 1.5$) and gapless ($J = 1.0$) cases, the error in the energy density becomes

independent of $n$ as $n \to \infty$, verifying the expectation in proposition 2. Furthermore, we show the error in the energy density for large $n$ as a function of $\delta$ in Fig. 2(b) and see that, consistent with proposition 2, this error $\to 0$ as $\delta \to 0$. Finally, Fig. 2(c) shows this error as a function of $J$—we see the error peak near $J = 1$ (i.e., the point where the gap in the Hamiltonian closes), and that it is smaller for values of $J$ where the model is gapped.

The translational invariance of the observables considered here is key to the stability result—translationally varying observables need not be stable, even if they are intensive and local. A simple example here is Anderson localization—consider $H$ to be a 1D translationally invariant tight-binding model i.e., $H = \sum_{i=1}^{n-1}(a_{i+1}^\dagger a_i + a_i^\dagger a_{i+1})$, with errors $\sum_{i=1}^{n} \delta_i a_i^\dagger a_i$ where $v_i$ is chosen uniformly at random between $[-\delta, \delta]$. The ground state of $H$ is completely delocalized across the spin-chain. In the presence of errors, no matter how small, this model is known to be localized. Now, for every $\delta_1, \delta_2...\delta_n$, consider the intensive observable $O_{\delta_1, \delta_2...\delta_n}$ given by the average particle numbers on $\Theta(1/\delta)$ sites around the site where the ground state is localized. This observable, when measured in the delocalized ground state of the unperturbed Hamiltonian $H$, yields an expected value of 0 as $n \to \infty$. On the other hand, in the ground state of the perturbed localized model it will yield an expected value of $\Theta(1)$. Thus, not all translationally varying observables can be stable, even if we restrict ourselves to Gaussian fermion models, with the observables being intensive and spatially local.

Next, we consider the Gibb's state, $e^{-\beta H}/\mathrm{Tr}(e^{-\beta H})$ where the inverse-temperature $\beta$ is a constant independent of $n$ and again study the stability of translationally invariant Gaussian observables that are generated $k-$locally. We show the following proposition in supplemental note IIC.

**Proposition 3.** The quantum simulation task of measuring translationally invariant Gaussian observables generated $k-$locally as described by Eq. (7), in the Gibbs state at inverse-temperature $\beta$ of a spatially local Gaussian fermion Hamiltonian is stable to coherent Hamiltonian errors as modeled by Eq. 4a with $f(\delta) = O(\beta\sqrt{\delta})$.

We point out that, in contrast to the corresponding result for ground states, this stability result corresponding to the Gibbs state does not rely on an assumption on the density of modes of the target Hamiltonian. However, $f(\delta)$ grows with $\beta$, so this result does not directly imply the stability of the ground state since $\beta$ would in general have to be increased with $n$ for the Gibbs state to approximate the ground state. However, for models with a gap above the ground state, as well as only very few low-energy eigenstates, at sufficiently low temperatures (i.e., high $\beta$), stability of the Gibb's state is essentially the stability of the ground state—in this case, the bound from proposition 2 should be applicable and would yield $f(\delta)$ independent of $\beta$.

In the more general setting of a Markovian open quantum system, the fixed point of the master equation would capture its equilibrium properties. Here, the quantum simulator is configured to implement the Hamiltonian and jump operators in Eq. 3 but instead, due to coherent and Markovian incoherent errors, implements the perturbed Hamiltonian and jump operators in Eq. 4. Similar to the case of the Hamiltonian ground state problem, we make an assumption on the spectral properties of the target Lindbladian—in particular, similar to assumption 1, we assume (Hölder) continuity of the fraction of modes with decay rates in the interval $[0, \eta]$,

**Assumption 2.** (Informal). The Gaussian Lindbladian has a unique fixed point and the number of eigenmodes $n_\eta$ with decay rates lying in the interval $(0, \eta]$, for sufficiently small $\eta$, satisfies the upper bound

$$n_\eta \leq nf_\ell(\eta) + \kappa(\eta, n), \tag{10}$$

where $n = DL^d$ is the number of fermionic modes, $f_\ell(\eta) \leq O(\eta^\alpha)$ as $\eta \to 0$ for some $\alpha > 0$ and $\kappa(\eta, n) \leq o(n)$ for any fixed $\eta$.

We provide a precise definition of eigenmode decay rate of a Gaussian Lindbladian in supplemental note IID. We remark that this assumption is very mild and is expected to be satisfied for physically relevant models—in particular, similar to the case of Gaussian Hamiltonians, this assumption is satisfied for translationally invariant models. Furthermore, this assumption is expected to be satisfied for Gaussian fermion models that are rapidly mixing in which case all the modes other than the fixed point mode typically have a system size independent decay rate (i.e., there is a gap in the decay rate spectrum of the Lindbladian)[38]. Beyond rapidly mixing Lindbladians, assumption 2 includes systems which have eigenmodes with decay rates scaling as $O(1/n)$ (i.e., the Lindbladian decay rate spectrum is gapless), and take a much longer time ($\sim\Theta(n)$) to reach its fixed point. For models satisfying assumption 2, we show the following proposition in supplemental note IIA.

**Proposition 4.** The quantum simulation task of measuring translationally invariant Gaussian observables generated $k-$locally as described by Eq. (7), in the fixed point of a spatially local Gaussian fermion Lindbladian whose distribution of modes satisfies Eq. (10) is stable to coherent and Markovian incoherent errors as modeled by Eqs. (4a) and (4b) with $f(\delta) = O(\delta^{1/2}) + O(f_\ell(\delta^{1/4})) \leq O(\delta^\beta)$ for some model-dependent constant $\beta$.

## Quantum spin systems

While for Gaussian fermion models, we could prove tight stability results with minimal assumptions on the model, looser stability results hold for quantum spin systems under more restrictive assumptions on their many-body spectrum. Here, we consider more general spin systems and show the stability of several quantum simulation tasks, in both dynamics and equilibrium, using locality results that have already been established in the many-body literature[39–46].

**Finite time dynamics.** Consider first the setting where an initial state $\rho(0) = |0\rangle\langle 0|^{\otimes n}$ is evolved under a Lindbladian $\mathcal{L}$ (Eq. (1)) for a time $t$ that is independent of $n$. Since geometrically local Lindbladians are expected to have a finite velocity of correlation propagation, observables at time $t$ should only be impacted by qubits which are within a $\sim t$ distance of their support and thus not be impacted errors on all the qubits. This motivates us to consider observables $O$ that are supported on $n-$independent number of qubits—more specifically, we consider observables that are either local (i.e., act non-trivially on an $n-$independent subset of spins), or of the form

$$O = O_1 O_2 \ldots O_k, \tag{11a}$$

where $O_1, O_2...O_k$ are local and $k$ is independent of $n$, or

$$O = \sum_{i=1}^{M} w_i O_i, \tag{11b}$$

where $\sum_{i=1}^{M} |w_i| = 1$, $O_i$ are of the form of Eq. 11 and $M$ can possibly grow with $n$. For these observables, the stability of this quantum simulation task can be stated:

**Proposition 5.** The quantum simulation task of measuring $k-$local observables, or their weighted averages, for constant-time dynamics under a spatially local Lindbladian is stable under coherent and incoherent errors with $f(\delta) = O(t^{d+1}\delta)$.

The proof of this result, provided in supplemental note IIIA uses the Lieb-Robinson bounds[39,40,43]. We note that a similar result has been proven for coherent errors and Markovian incoherent errors in ref. 38. Our contribution is to show that this bound holds in the more general setting of coherent and non-Markovian incoherent errors, and thus is

more directly applicable to experimentally realistic quantum simulators.

Note also that for large $t$, the error between the target observable and the observable measured on the quantum simulator grows as $t^{d+1}$—this result can be interpreted by noting that for a geometrically local model in $d$-dimensions, the number of qubits within the light-cone of a local observable at time $t$ is $~t^d$—since any error in a qubit within the light cone and at any time $t$ could lead to an error in the local observable, we obtain a worst-case bound scaling as $~t^{d+1}$ in the observable value. Furthermore, this is looser than the corresponding result in Gaussian fermion models (proposition 1), where the error grows only as $t$.

**Equilibrium.** We next study the stability of the task of simulating the ground state and Gibbs states of $H$, and focus only on understanding the impact of coherent Hamiltonian errors. This problem has been previously extensively investigated in many-body physics[41,45,46], and in this section we cross-examine these results from the perspective of analog quantum simulation. We assume that $H$ is gapped i.e., the energy difference between the ground state and the first excited state is larger than a constant $\Delta$ independent of $n$. Furthermore, we also assume that the Hamiltonian remains gapped in the presence of errors —we refer to such a target Hamiltonian $H$ to be stably gapped. We point out that the stability of the gap in the presence of errors or perturbations has only been shown for certain frustration free models with local topological order[47–50] and is not true for all gapped models[49,51–53], but we posit it as a reasonable physical assumption. The stability of this quantum simulation task is a direct consequence of the spectral flow method developed by Hastings et al.[41,45] which shows that there exists a unitary taking the ground state of $H$ to the ground state of $H'$ that is quasi-local. We thus obtain the following proposition and we include a proof of this in supplemental note IIIB.

**Proposition 6.** (From refs. [41,54]). The quantum simulation task of measuring $k$-local observables or their weighted averages (Eq. 11), in the ground state of stably gapped spatially local Hamiltonians is stable to coherent Hamiltonian errors with $f(\delta) = O(\delta)$.

The choice of observables here is crucial to having a stable quantum simulation task—it is well understood that even for stably gapped Hamiltonians, non-local observables would not be stable. Furthermore, we point out that for the case of Gaussian Fermions and for translationally invariant local observables, our stability result (proposition 2) is less restrictive—in particular, it does not require even the existence of a gap in the target Hamiltonian.

We next consider the Gibbs state of $H$ at some temperature $\beta$ independent of $n$, and assume that the Gibbs state has an exponential clustering of correlation[46] i.e., for any two observables $A$, $B$ separated by distance $l$,

$$|\langle A \otimes B \rangle - \langle A \rangle \langle B \rangle| \le \|A\|\|B\|O(e^{-c_2 l}),$$

for some model-dependent constant $c_2$. Furthermore, as in the case of ground states, we assume that this exponential clustering of correlations is stable under errors. In this case, we obtain that the problem of measuring $1$−local observables and their weighted averages is stable—a proof of this is included in supplemental note IIIC.

**Proposition 7.** (From ref. [46]). The quantum simulation task of measuring local observables or their weighted averages, as given by Eq. 11 with $k = 1$, in the Gibbs state of spatially local Hamiltonians with stable exponential clustering of correlations is stable to coherent Hamiltonian errors with $f(\delta) = O(\delta \log^d(1/\delta))$.

More generally, we can consider the problem of finding local observables in the fixed points of spatially local Lindbladians. For this, we restrict ourselves to rapidly mixing Lindbladians which were identified in ref. [38]—a Lindbladian $\mathcal{L}$ on $n$ spins with fixed point $\sigma$ is rapidly mixing if $\|e^{\mathcal{L}t} - \mathrm{Tr}(\cdot)\sigma\|_\diamond \le \mathrm{poly}(n)e^{-\Theta(t)}$ i.e., irrespective of the initial state, the state of the system converges exponentially fast to the fixed point $\sigma$. Under this assumption, we can show the following stability result.

**Proposition 8.** The quantum simulation task of measuring $k$-local observables, or their weighted averages, in the fixed point of spatially local Lindbladians which satisfy rapid mixing is stable to coherent and incoherent errors with $f(\delta) = O(\delta)$.

The proof of this proposition, presented in supplemental note IIID, builds on the analysis in ref. [38]—in particular, we point out that ref. [38] already establishes that local observables in fixed points of spatially local rapidly mixing Lindbladians is stable to coherent errors and incoherent Markovian noise. Our key contribution is to extend this to the more general and experimentally realistic setting of non-Markovian noise.

## Quantum advantage without noise

In the absence of any noise, the advantage of a quantum algorithm over a classical algorithm is often formulated in terms of their run-time scaling with respect to the system size. However, in many-body physics problems, the quantities of interest are the value of certain intensive observables in the thermodynamic limit i.e., when the system size $n \to \infty$. Consequently, it is less meaningful to consider the algorithm's complexity as a function of system size and instead consider it as a function of the target precision $\varepsilon$ demanded in the computed thermodynamic limit[35,36]. Apart from being theoretically meaningful, expressing run-times in terms of precision might additionally be practically relevant in scenarios where we are trying to calculate either phase transition points[53,55] or critical exponents characterizing a phase transition, both of which are typical calculations of interest in many-body physics. More precisely, let us consider a many-body model defined as a family of Lindbladians $\{\mathcal{L}_n\}_{n\in\mathbb{N}}$ and observables $\{O_n\}_{n\in\mathbb{N}}$, where $\mathcal{L}_n, O_n$ act on $n$−spins. We are interested in the expected value of $O_n$ in a many body quantum state, e.g., in equilibrium or dynamics, associated with the Lindbladian $\mathcal{L}_n, \rho_{\mathcal{L}_n}$. We furthermore assume that the models and observables under consideration have a well-defined thermodynamic limit i.e.,

$$O^* := \lim_{n\to\infty} \mathrm{Tr}\left(\rho_{\mathcal{L}_n} O_n\right) \tag{12}$$

exists. Now, given a precision $\varepsilon$, it is reasonable to assume that we can then choose $n$ as a function of $\varepsilon$ such that

$$|O^* - \mathrm{Tr}(O_n \rho_{\mathcal{L}_n})| \le \varepsilon, \tag{13}$$

i.e., approximate the thermodynamic limit by a finite-size problem (note however, one can artificially construct models where this may not be strictly possible[53]). The run-time of a quantum simulation or a classical simulation for the finite-size problem can thus be expressed in terms of the precision $\varepsilon$ demanded in the thermodynamic limit. This allows us to then compare the scalings of the run-time of these algorithms with the precision $\varepsilon$, and declare an algorithm to have an advantage in precision compared to others depending on their respective scaling. For instance, if a quantum algorithm has a complexity $T_Q = \mathrm{poly}(1/\varepsilon)$, then we will have a superpolynomial advantage over a classical algorithm with complexity $T_{cl} = \exp(\log^2(1/\varepsilon))$ and exponential advantage over a classical algorithm with complexity $T_{cl} = \exp(O(1/\varepsilon))$.

**Finite-time quantum dynamics.** We consider first an initial product state $|0\rangle^{\otimes n}$, and for $t > 0$, we take $\rho_{\mathcal{L}_n} = e^{\mathcal{L}_n t}((|0\rangle\langle 0|)^{\otimes n})$. The observable of interest is a fixed local observable $O_n = O$. The existence of the thermodynamic limit is obtained directly using the Lieb-Robinson

bounds[39,40], which also characterize the error between the thermodynamic limit and its finite-size approximation—for the problem of computing a local observable after evolving $|0\rangle^{\otimes n}$ for finite-time $t$ with respect to a $d$−dimensional spatially-local Hamiltonian, $O^*$ exists and fulfills Eq. (13) for $n = \Omega(\log^d(1/\varepsilon) + t^d)$. On an ideal quantum simulator, one would evolve $n = \Theta(\log^d(1/\varepsilon) + t^d)$ qubits for a time $t$ and measure the observable. The procedure would be repeated $\Theta(1/\varepsilon^2)$ to reduce the measurement error in the observable to $\varepsilon$, yielding a total run-time of $O(t/\varepsilon^2)$.

On a classical computer, in general, the only algorithms with a rigorous guarantee known for the problem of computing local observables for a finite number of spins $n$ are either exact diagonalization or Krylov subspace methods[56]. Using either of these methods has a worst-case run-time that scales at least exponentially with the number of spins $n$. To compute the thermodynamic limit to a precision $\varepsilon$, we would first approximate the thermodynamic limit by a finite-size problem and then use these classical algorithms on the resulting finite-size problem. In supplemental note IVA (proposition 1), we exhibit a local observable in a simple nearest-neighbor tight-binding model on $\mathbb{Z}^d$ such that the system size needed to approximate its thermodynamic limit to a precision $\varepsilon$, for a fixed $t$, is at least $\Omega(\log^d(\Theta(\varepsilon^{-1}))/\log^d\log(\Theta(\varepsilon^{-1})))$. Thus, exact diagonalization or Krylov subspace methods scale at least super polynomially with $\varepsilon^{-1}$ on worst-case instances, yielding a super polynomial advantage of using quantum simulators.

We note that there are several classical heuristic algorithms, which use an efficiently contractable tensor network ansatz, that in many problems are much faster than the worst case[57,58]. However, these methods do not have rigorous guarantees—since, when noiseless, the quantum simulation of dynamics is not a heuristic, we only compare it to classical algorithms with rigorous guarantees. We also point out that one could also use other classical methods that operate directly in the thermodynamic limit instead of analyzing a finite-size approximation[59,60]. For instance, in ref. 59 a method based on cluster expansion was analyzed for which the computational time is upper bounded by poly($\varepsilon^{-1}$), although this upper bound scales super-exponentially with time. To the best of our knowledge, providing a lower bound on this classical algorithm is an open problem. However, assuming that it has the same scaling with $t$ as the upper bound provided in ref. 59, a quantum simulator will have an exponential quantum advantage with respect to this method for evolution times $t \sim$ poly($\varepsilon^{-1}$).

**Ground state.** Consider next the problem of estimating local observables in the ground state of many-body Hamiltonians in the thermodynamic limit. The convergence rate of a finite-size approximation of a local observable to its thermodynamic limit for the ground state problem is expected to depend on whether the model is gapped (and hence the ground state has exponentially decaying correlations[42,61]) or gapless. While it is generally hard to rigorously characterize the rate of convergence of a finite-size approximation to the thermodynamic limit for ground states, it is physically reasonable to assume either Logarithmic Convergence, where Eq. (13) holds for $n = \Omega(\log^d(1/\varepsilon))$ with $d$ being the lattice dimension, or Power-Law Convergence where Eq. (13) holds for $n = \Omega(\text{poly}(\varepsilon^{-1}))$. The first case is expected to hold for gapped models, and can be rigorously established for models satisfying local topological quantum order condition[47,49,50]. Additionally, for gapped models, Logarithmic Convergence is expected to be tight since we can easily construct examples of spatially local gapped Hamiltonians (such as the AKLT model[62]), for which a system-size of at least $\Omega(\log^d(\varepsilon^{-1}))$ is needed to approximate the thermodynamic limit of a local observable to a precision $\varepsilon$ (see supplemental note IVB, proposition 2). Power-Law Convergence is expected to hold for critical (gapless) models—for instance, this is the case for the Gaussian fermionic Hamiltonians analyzed in the previous section, under very general conditions for the Fermi surface. Similar to the situation with dynamics, currently

available classical algorithms with rigorous guarantees to compute a ground state observable use either exact diagonalization or a Krylov subspace method on the finite-size Hamiltonian approximating the thermodynamic limit. Thus, for models satisfying the Logarithmic Convergence condition, a classical computer would require time $\exp(\Omega(\log^d(\varepsilon^{-1})))$ in the worst case. Instead, for models satisfying the Power-Law Convergence condition, a classical computer is expected to require time $\exp(\Omega(\text{poly}(\varepsilon^{-1})))$.

Furthermore, to ensure that there is a quantum algorithm that reaches the ground state we will assume that $H_n$ is adiabatically connected to a family of Hamiltonians $H_n^{(0)}$ with efficiently preparable ground states such that the minimal gap, $\Delta_n$, along the adiabatic path fulfills $\Delta_n \geq \Omega(1/\text{poly}(n))$. This assumption ensures that using the adiabatic algorithm one can reach the ground state within an error $\varepsilon$ in a time $T_Q = \text{poly}(n, 1/\varepsilon)$, or $T_Q = \text{poly}(1/\varepsilon)$ if framed entirely in terms of the precision of the thermodynamic limit and is expected to hold for physically relevant gapped or gapless models. Moreover, for a constant gap, and with certain further assumptions on the frustration of the Hamiltonian, it is provably possible to reach the ground state in $T_Q = \text{polylog}(n, 1/\varepsilon)$[63]. Comparing this run-time with those of the classical algorithms discussed above, we then expect a superpolynomial quantum advantage for (gapped) models satisfying Logarithmic Convergence to the thermodynamic limit, and (gapless) models satisfying Power-Law Convergence to the thermodynamic limit.

**Fixed points.** Similar to ground states of many-body Hamiltonians, we can consider the problem of computing local observables in the fixed points of many-body Lindbladians. Depending on the spectral properties of the Lindbladian, a local observable in the fixed point may exhibit Logarithmic or Power-Law Convergence to the thermodynamic limit. In particular, for rapidly mixing Lindbladians, it has been shown in ref. 38 that local observables in the fixed point exhibit a logarithmic convergence to the thermodynamic limit. Additionally, this convergence is tight i.e., we can exhibit a specific rapidly mixing Lindbladian for which a system-size of at least $\Omega(\log^d(\varepsilon^{-1}))$ is needed to approximate the thermodynamic limit of a local observable to a precision $\varepsilon$ (see supplemental note IIIC, proposition 3). Furthermore, Lindbladians which are not rapidly mixing but take time polynomial in the system size to reach their fixed points would have local observables satisfying Power-Law Convergence. Examples of such Lindbladians would be those corresponding to Glauber dynamics corresponding to the 2D critical Ising model[64].

As with the ground state problem, Krylov subspace or exact methods, which have a rigorous guarantee, would require a worst-case time $\exp(\Omega(\log^d(\varepsilon^{-1})))$ for models with logarithmic convergence and $\exp(\Omega(\text{poly}(\varepsilon^{-1})))$ for models with power-law convergence. Furthermore, under the physically-motivated assumption that Lindbladian dynamics, for a finite system with system-size $n$, reaches its fixed point in at most poly($n$) time, an ideal quantum simulator that implements this Lindbladian would require a time $O(\text{poly}(1/\varepsilon))$ to approximate the thermodynamic limit of local observable to a precision $\varepsilon$. Thus, we expect to obtain a superpolynomial quantum advantage for problems with logarithmic convergence, and exponential quantum advantage for problems with power-law convergence.

## Quantum advantage with noise

In the presence of errors, the arguments formulated in the previous subsection no longer hold. For unstable quantum simulation tasks, we expect thermodynamic limits in the presence of errors to be a bad approximation to the target thermodynamic limit. However, for stable quantum simulation tasks in particular, the noisy quantum simulator can still produce a faithful approximation of the thermodynamic limit with the hardware error $\delta$ setting a limit on the obtained precision. More precisely and as depicted in Fig. 3, in time poly($1/\varepsilon'$), the quantum simulator is expected to compute the thermodynamic limit of the

noisy model to a precision $\varepsilon'$—the precision of the target thermo-dynamic limit obtained is thus upper bounded by

$$\varepsilon \le O(\max(\varepsilon', f(\delta))),$$

where $f(\delta)$ is given in Definition 1. Therefore, in the presence of hardware errors, the quantum simulator need not be run beyond a time needed to obtain $\varepsilon' = f(\delta)$, and we can expect to compute the target thermodynamic limit to a precision of $O(f(\delta))$. As summarized in Table 1, we typically obtain $f(\delta) = O(\delta^\alpha)$ for $\alpha > 0$ for most stable many-body simulation tasks, and thus to be able to obtain the thermodynamic limit to a precision of $O(\text{poly}(\delta))$, determined entirely by the hardware error $\delta$, in quantum-simulation time $O(\text{poly}(1/\delta))$.

A numerical illustration of this analysis is shown in Fig. 4—here, we use the adiabatic quantum algorithm to find the energy density observable in the ground state of the critical SSH model (i.e., Eq. (9) with $J = 1$). Figure 4a shows the convergence of the energy density observable, in the absence of errors, to its thermodynamic limit—we see that a power-law convergence is obtained, as physically expected for gapless models. In Fig. 4b, we use a system-size that yields a precision of $O(f(\delta))$, as determined by the stability bounds on the ground-state of this model, and simulate an adiabatic algorithm to find the ground state in the presence of hardware error. We see that, in the presence of errors, the accuracy in the achieved precision is fundamentally limited by the hardware precision $\delta$—Fig. 4 shows the run-time of the adiabatic algorithm as a function of this hardware-limited precision. We see that this run-time scales polynomially with $1/\varepsilon$, where $\varepsilon$ is the hardware-limited precision that is achieved by the adiabatic algorithm.

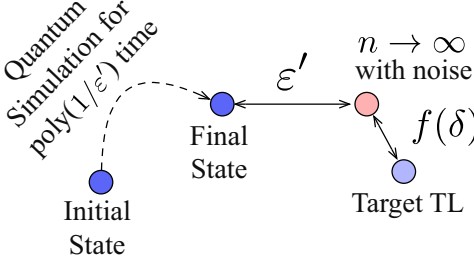

**Fig. 3 | Precision and run-time attained by noisy quantum simulators.** An erroneous quantum simulator can obtain the thermodynamic limit of the perturbed model to a precision $\varepsilon'$ in time $\text{poly}(1/\varepsilon')$—this thermodynamic limit, however, can have an error $f(\delta)$ from the target thermodynamic limit in the presence of hardware error $\delta$.

To define a notion of advantage in the presence of noise, we can now compare the classical and quantum run-times needed to achieve this hardware-limited precision. Assuming $f(\delta) = \text{poly}(\delta)$, it follows from the discussion in the previous subsection that we would need classical run-times that are either superpolynomial or exponential in $\text{poly}(1/\delta)$ to achieve the precision that can be achieved by quantum simulators in time $\text{poly}(1/\delta)$. That is, if $\delta$ is decreased by a constant factor, then the run-time of the quantum simulator will only increase at-most polynomially with this factor, while the run-time of the classical simulator will increase by a super-polynomial or exponential factor. We summarize our expectation of noisy quantum advantage for the quantum simulation task of finding local observables in dynamics, ground states and fixed points in the Table 2.

We emphasize the following two points regarding Table 2. First, the "provable" quantum advantage for dynamics, stably gapped ground states with logarithmic convergence, and fixed points of rapidly mixing Lindbladians is only with respect to Krylov subspace methods or exact methods, which are the only known classical algorithms with rigorous guarantees known for these models. To the best of our knowledge, it remains an open problem to provide a universal lower bound on any possible classical algorithm or even connect a possible quantum advantage for the specific many-body problems that we consider to well-known complexity assumptions. Second, we only conjecture the quantum advantage for critical models (both ground states and fixed points)—our conjecture is based on the novel stability results that we provide for Gaussian fermion models and under the expectation that these models could capture the qualitative physics of more complex non-Gaussian models, even though Gaussian fermion models on their own can be solved efficiently on classical computers.

## Discussion

We have considered both the stability and quantum advantage of using near-term analog quantum simulators for thermodynamic limits of many-body problems in physics. Based on both existing theoretical results in many-body literature, and new technical results for Gaussian fermion models, we argue that many physically relevant many-body problems are stable to a constant rate of error on the quantum hardware being used to solve them and thus are accessible in near-term experiments. We also hypothesize that these algorithms have an advantage, with respect to the obtained precision, in computing thermodynamic limits of many-body problems. Our formulation and results provides some evidence for near-term analog quantum simulators being useful for solving many-body problems.

Extending the stability results for gapless/critical models to the case of quantum spins, or non-Gaussian fermionic systems is

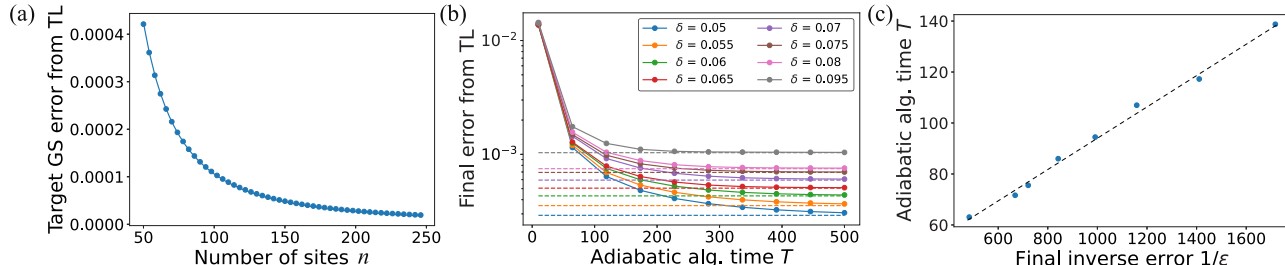

**Fig. 4 | Numerical study of quantum adiabatic algorithm in the presence of error.** We consider using the adiabatic algorithm to find the energy density observable for the critical SSH model in the thermodynamic limit (TL).
**a** Convergence of the energy density to the thermodynamic limit as $n \to \infty$—the scaling of $\varepsilon$ with $n$ reveals a power-law scaling that is expected for gapless models. **b** The adiabatic algorithm in the presence of hardware errors—the quantity being plotted is the error of the noisy adiabatic algorithm from the thermodynamic limit of the noiseless model. The precision achieved by the adiabatic algorithm is fundamentally limited by hardware errors. **c** The final precision ($\varepsilon$) achieved by an adiabatic algorithm in the presence of errors as a function of the adiabatic algorithm run-time, confirming that $T \sim \text{poly}(1/\varepsilon)$ as expected from our analysis. Thus, on decreasing the hardware error, the error achievable by the noisy quantum simulator decreases and the run-time of the quantum algorithm increases at most polynomially.

**Table 2 | Summary of the classical and quantum run-times for thermodynamic limits needed to obtain the hardware-limited precision with hardware error/noise δ**

| Problem | Error model | Assumption | Classical run-time | Quantum run-time |
|---|---|---|---|---|
| *Provable noisy quantum advantage over Krylov Subspace/Exact methods* | | | | |
| Dynamics | General errors | None | $\exp(\tilde{\Omega}(\log^d(\Theta(\delta^{-1}))))$ | $O(\text{poly}(\delta^{-1}))$ |
| Ground state | Coherent | • Stable gap | $\exp(\Omega(\log^d(\Theta(\delta^{-1}))))$ | $O(\text{poly}(\delta^{-1}))$ |
| | Hamiltonian | • Logarithmic convergence | | |
| | errors | • Adiabatic path with a gap $> \Omega(1/\text{poly}(n))$ | | |
| Fixed points | General errors | Rapid mixing | $\exp(\Omega(\log^d(\Theta(\delta^{-1}))))$ | $O(\text{poly}(\delta^{-1}))$ |
| *Conjectured noisy quantum advantage* | | | | |
| Ground states | Coherent | • Stable $\Omega(1/\text{poly}(n))$ gap | $\exp(\Omega(\text{poly}(\delta^{-1})))$ | $O(\text{poly}(\delta^{-1}))$ |
| | Hamiltonian | • Power-law convergence | | |
| | errors | • Adiabatic path with a gap $> \Omega(1/\text{poly}(n))$ | | |
| Fixed points | Coherent and incoherent Markovian errors | • Reaches $\varepsilon$—close to fixed point in $O(\text{poly}(n, 1/\varepsilon))$ time <br> • Power-law convergence | $\exp(\Omega(\text{poly}(\delta^{-1})))$ | $O(\text{poly}(\delta^{-1}))$ |

For classical run-times, we only consider Krylov subspace methods or exact diagonalization as the classical algorithm for the provided lower bounds, and not heuristics which do not have rigorous convergence guarantees. Note that $\tilde{\Omega}$ suppresses $\log\log(\delta^{-1})$ factors.

an immediate open problem suggested by our work. While previous work by Hastings[41] indicates that, under some assumption on the density of states of the many-body model, such a stability result could hold for gapless spin systems, it would be interesting to see if restricting observables to being translationally invariant could help improve these results. Similarly, understanding the stability of Lindbladian dynamics and fixed point problems for quantum spin systems or non-Gaussian fermionic systems beyond the rapid mixing assumption and under both Markovian and non-Markovian errors would also be important directions. This has been under investigation recently in the context of defining stable phases of matter[65,66].

## Methods

Here, we provide the key ideas behind the proofs of the theoretical results presented in results section. Detailed proofs building upon the ideas presented in this section can be found in the supplement.

To prove the stability results for the Gaussian fermion problems, we utilize the fact that both equilibrium and dynamical properties of Gaussian fermion models can be captured by just the covariance matrix of the state at hand—more specifically, for a time-dependent Gaussian state $\rho(t)$, the corresponding covariance matrix is given by

$$(\Gamma(t))_{x,y}^{\alpha,\beta} = \frac{1}{2}\text{Tr}\left(\left[c_x^\alpha, c_y^\beta\right]\rho(t)\right),$$

and its dynamics, for the quantum simulator model with gaussian errors and noise sources described in results section, is derived in lemma 3 in supplemental note II. The stability result for dynamics (proposition 1) is obtained from an application of rigorous time-dependent perturbation theory on the effective equations of motion for the covariance matrix.

For analyzing stability of equilibrium states, as described in the results section (propositions 2–4), we consider only translationally invariant observables (Eq. (7)). Suppose $\rho_0$ and $\rho_\delta$ respectively denote the noiseless and noisy equilibrium Gaussian states, then in lemma 6 of supplemental note II we establish the following bound on the error in the expected translationally invariant observable $O$

$$|\text{Tr}(O\rho) - \text{Tr}(O\rho_\delta)| \le \frac{4D^2 k}{n}\left|\left|\tilde{O}_0\right|\right|\left|\left|\Gamma - \Gamma_\delta\right|\right|_1, \quad (14)$$

where $\Gamma$ and $\Gamma_\delta$ are the covariance matrices corresponding to $\rho$ and $\rho_\delta$ respectively, $O_0$ is a $k$—local observable defined in Eq. (7) and $\tilde{O}_0$ is a $2n \times 2n$ matrix with matrix elements $o_{x,y}^{\alpha,\beta}$ such that

$$O_0 = \sum_{x,y \in \mathbb{Z}_L^d} \sum_{\alpha,\beta=1}^{2D} o_{x,y}^{\alpha,\beta} c_x^\alpha c_y^\beta.$$

We remark that the bound in Eq. (14) is obtained by explicitly utilizing the translational invariance of $O$ and hence its block-diagonality in the momentum basis. Without translational invariance, the best bound on $|\text{Tr}(O\rho) - \text{Tr}(O\rho_\delta)|$ would be $||\tilde{O}_0||||\Gamma - \Gamma_\delta||_1$, which for $k, D \le O(1)$ would asymptotically be much worse than the bound in Eq. (14). Building on Eq. (14), in supplemental note II, we then show that $||\Gamma - \Gamma_\delta||_1 \le O(n\delta^\alpha)$, for some $\alpha > 0$, for all the considered equilibrium problems (ground state, Gibb's state, fixed points).

The proofs of stability results for spin systems (propositions 5–8) provided in supplemental note III builds on the quasi-locality results in refs. [39–46]. Proofs of propositions 5 and 8 build upon[38] and additionally consider non-Markovian perturbation to the dynamics. The key technical ingredient to handle non-Markovian perturbations is the input-output equations for the environment which are commonly used in quantum optics[67]. The input–output formalism allows us to effectively locally bound the effect of non-Markovian perturbations on the dynamics of the quantum simulator, and together with Lieb-Robinson bounds[43] allow us to prove propositions 5 and 8. The proofs of propositions 6 and 7 are only minor modifications of the quasi-locality results in ref. [45] and ref. [46] respectively.

## Code availability

The code used for generating numerical results in this paper is publicly available[68].

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

## Acknowledgements

We acknowledge useful discussions from Dorit Aharonov and Álvaro M. Alhambra. This research is funded by the German Federal Ministry of Education and Research (BMBF) through EQUAHUMO (Grant No. 13N16066) within the funding program quantum technologies—from basic research to market and by the Munich Quantum Valley (MQV), which is supported by the Bavarian state government with funds from the Hightech Agenda Bayern Plus. R.T. also acknowledges funding from the Max-Planck Harvard Research Center for Quantum Optics (MPHQ) postdoctoral fellowship, as well as startup funding from University of Washington. A.F.R. is supported by the Alexander von Humboldt Foundation through a postdoctoral fellowship.

## Author contributions

R.T., A.F.R., and J.I.C. formulated the problem and developed proof ideas. R.T. and A.F.R. developed detailed calculations for the proofs and J.I.C. supervised the project. R.T. and A.F.R. contributed equally to the manuscript. All authors contributed to the writing of the manuscript.

## Funding

## Competing interests

The authors declare no competing interests.
