## [Peer Review File · Nature Communications]

Quantum advantage and stability to errors in analogue quantum simulatorsREVIEWER COMMENTS

Reviewer #1 (Remarks to the Author):

This is a report on the manuscript entitled "Quantum advantage and stability to errors in analogue quantum simulators". This work is addressing a timely question in quantum information science: It asks in what precise way we can hope analogue quantum simulators to have predictive power. At the heart of the matter is the issue of how errors in such prescriptions will build up, and what errors do to the quantum simulation.

After all, if they modify the results of the quantum simulation much, then there is little predictive power. Quantum error correction is out of scope for analogue quantum simulators, not only now, but presumably as a feature of such devices, as one deliberately tries to have high levels of control over the constituents, but does not aim at actually implementing quantum gates. Hence the stability to errors is a core research question in analogue quantum simulators.

The present work addresses this question. It is not the first manuscript discussing this question, but it adds a number of interesting technical points to it. On the highest level, the present work makes the point that errors tend to average away in a more beneficial fashion compared to quantum gate based quantum computing. The type of stability considered is a stringent one, namely a system-size independent notion of stability against extensive errors. This is a very stringent measure indeed, and it is not easy to prove such a stability for general quantum systems. For a very simple family of models, namely non-interacting or "Gaussian" settings, the stability in this sense is shown (and similar results are automatically inherited for some spin models).

This can be seen as an indication that this type of stability is realistic and is a solid start. What is interesting is that the criticality does not seem to matter much in the robustness, which may be seen as being surprising, in the light of the fact that correlations are long-ranged. It is further argued that this type of stability may lead to a quantum advantage, as it may indicate that this sort of stability is providing a kind of robustness that allows to make stable predictions, while classical simulation methods are more limited in time and system

size.

I see the biggest strength of the manuscript being its conceptual clarity and the definition of the stringent robustness measure. The technical results reflect solid work, but may not be too surprising. The strongest limitation could well be the ansatz class considered, in that non-interacting or "free" or Gaussian models are considered for a substantial proportion of the manuscript, while for general systems, stronger assumptions are being made. Within the free fermionic framework, even open analogue quantum simulations can be considered, in which two types of noise compete with each other. It would have been better, possibly, to not say that this model is chosen "to be concrete", as it is said. But because the setting is simple in this way, and since one can make strong predictions, this setting may still provide a good indication of what the general setting would look like.

Definition 1 is the core definition of this work: It may be a good idea to state it in the trace distance, as this is basically what this definition is about. Maybe more seriously, it should be a bit more precisely stated, as the operator norm of O is this way not taken into account. There is still a kind of "strength" of the observable, which should be taken into account on the right hand side. This is an omission, also because earlier, the diamond norm (as a stabilized channel trace norm) is made use of, without mentioning and defining it.

Proposition 1 is interesting and insightful. And still, when it is said that the dynamics is stable to coherent and incoherent errors, it should be stated what kind of Gaussian errors this is stable against.

Assumption 1 is also very reasonable. While this is not proven for specific models, I agree that this is a reasonable assumption. It would be good to relate it to more commonly used assumptions, say, on energy spacings of non-integrable models. I also understand that this assumption includes gapped models. What is less clear is what kind of critical models are allowed for.

Proposition 2 is then a bit unspecific: It would be good to be clear here also about the precise assumptions on the noise. Then, observables that are generated k -locally are

introduced in passing: But they are very important for the type of robustness encountered, and maybe it would be good to spend a bit more time on their physical interpretation. Proposition 3 on Gibbs state preparation is equally interesting, but it would be good to specify the underlying assumptions a bit more precisely. In Proposition 3 it is clear that one cannot naively apply the results on ground states. But for gapped models with not too many spectral values above the gap one could presumably still draw this conclusion, so it may be nice to specify the connection between Proposition 2 and 3 a bit more. Proposition 4 seems incomplete: What does "in the fixed point of a spatially local free-fermion" mean?

In Eq. (9), a different family of observables is considered. While this makes perfect sense from the perspective of the proof techniques explained in the appendix, it would be good to motivate this a bit more in the main text. The different dependence on the scaling in time and the dimension for spin models, can this be physically motivated?

To assume that the target Hamiltonian is stably gapped seems like a very strong assumption. This is indeed known for certain frustration free models with local topological order, but the perturbations allowed then are tiny. Would a relative perturbation also work?

For Gibbs states having an exponential clustering of correlations (an s is missing here), this would basically mean that the temperature is high enough, right? The reference to 1-local observables is a bit curious, however. I tried to derive this myself without looking at the appendix and did not see where the 1-locality assumption would come from. It might be good to explain better.

All this, of course, is the precursor of the discussion of presumably achieving a quantum advantage with analogue quantum simulators. This discussion starts out with discussing the ideal case - that of having no errors at all. This discussion of exact diagonalization techniques and Krylov subspace methods is insightful. The distinction between logarithmic convergence and power-law convergence is meaningful. What really matters, needless to say, is how noisy quantum simulators fare, and here the discussion becomes more vague. What follows is still a highly mature and insightful discussion: This is what we need to come closer to achieving realistic quantum advantages with analogue quantum simulators.

The results are valid, original and of significance. There is a strong emphasis on problems that allow for a relatively simple solution, but this is due to the complexity of the problem. The approach taken is nevertheless expected to be helpful. I most enjoyed going through the appendices. While the results are not overly technically involved, I would think that the manuscript constitutes a solid discussion document. This is helped by the good table providing an overview. I hence tend to recommend publication.

Reviewer #2 (Remarks to the Author):

The main aim of this paper is to study noisy quantum simulators to compute or approximate some relevant physical properties of many-body systems. These properties are studied for Gaussian fermions and for some spin models, and both in the cases of equilibrium and undergoing dynamics.

In the main text of the paper, the authors present a detailed discussion on the state of the art of analog quantum simulators, both motivating their use in various contexts, for different systems, as well as enlisting some of the main results for these objects the past few years. In particular, they discuss why a systematic and fundamental study of the quantum advantage for quantum simulators presents nowadays several theoretical issues, motivating the necessity to modify the standard definition in this framework. For that, they shift the study of many-body observables as a function of the system size to the intensive case, to end up showing that quantum simulators from some critical and non critical models are stable against errors.

First, they describe the notion of stability to consider in their quantum simulation tasks. Next, they address the case of Gaussian fermion models, for which they provide results both in the cases of finite-time dynamics and equilibrium. For the former, they show that measuring local Gaussian observables is stable to coherent and incoherent errors, whereas for the latter they study the analogous task measuring in the ground state and Gibbs state, respectively, showing stability with respect to coherent errors. This is subsequently extended to fixed points of rapidly mixing Lindbladians, allowing for coherent and Markovian incoherent errors.

In the next section, they present a collection of results for quantum spin systems, again both in the contexts of finite-time dynamics and equilibrium. The results shown are analogous to those for fermions, with the caveat that more restrictive assumptions have to be considered in the model. They conclude with a detailed section on quantum advantage with noisy quantum simulators, where they compare the ideal scenario with the noisy one.

Overall, I find the paper to be a very important contribution to the development of the field of quantum simulators, and I especially appreciate its careful writing. The main part of the manuscript is very enjoyable to read and the results are presented in a really clear way. The tables notably contribute to the understandability of the paper, and help the reader compare the results for both sets of models. For all this, I think that the paper should eventually be published in Nature Communications. However, before recommending acceptance, I have a few concerns about certain aspects of the proofs contained in the appendices that should be clarified by the authors.

Major concerns:

- My main concern appear in Section III of the appendix, namely the “Stability of spin models”. In Lemma 12, they consider a family of operators $E'(s,t)$ and show an identity concerning them, as well as the bosonic annihilation operators from the Lindbladian and the memory kernel. However, to prove it, they consider a superoperator which in particular contains $(E')^{(-1)}$. Why is this operator invertible? There are many examples in the literature in which this is a very subtle matter (e.g. in the context of adiabatic theorems for Lindbladians). Are the authors assuming anything in particular for L' so that the invertibility is guaranteed? This is not clear to me, and seems to be essential for the proof of this lemma and also for the rest of the section. One possibility is that they are only considering the “inverse” of the time-ordering, namely the reverse time ordering operator. In such a case, I don't see either how to obtain the derivative in the next line, since the A operators shouldn't commute with L' in general, but at least everything would be well defined. Could you please explain this point?

- On Page 36, in the proof of the upper bound for $e_{\{\alpha, \>\}}$, I find the whole argument quite confusing. First, after applying Lemma 19, I cannot see why you obtain Tr_E inside the norm of the second term. Secondly, the first term of that summand seems to disappear in the next line, and I cannot see why it should vanish without further explanation. Next, the last inequality just consists of adding positive terms, that are also not justified. It would be great if you could give a further intuition/explanation of what is going on here.
- On Page 25, the equation after 8 is really confusing to me. First, if $t(x)$ is defined from the \tanh , with domain in \mathbb{R} , why do you need a 2π -periodic extension of it? Next, with this definition, I don't understand the last equality in the equation after 8. How can the expression with $t(x-y)$ and $t(x+y)$ coincide? Is it due to the product with the Dirichlet kernel? This is not immediately obvious to me from its expression.

Minor concerns:

- In the notational preliminaries, you should mention the meaning of a norm without any subindex, since this is used quite frequently later.
- On Page 2, definition 1, doesn't f need to be continuous or monotone, at least for n large enough?
- On Page 20, the proof of Proposition 1 forces the reader to look for the definition of O in the main text. Maybe you could recall it here, so that it's easier to compare it to \tilde{O} .
- On Page 20, the proof of Lemma 6 is slightly confusing with the change to the vectorized matrices and back. Maybe you could simplify it by directly working with the blocks and their norms. But the proof is perfectly fine like this, so of course feel free to leave it like this.
- On Page 20, Equation 2, should it be an inequality? Otherwise it's a really strong assumption.
- Everywhere in Appendix III, and from hereafter, the numbering of the equations is different from that of the previous Appendix for fermions.
- Page 39. The proof of proposition 9 is a bit confusing because of the order. It was mentioned that you would compute a lower bound for E_n two pages before, but it's only mentioned here towards the middle of the proof, and not in the statement of the proposition. If you recall it here, it's better for the understanding. It would also be better for the reader if you mentioned explicitly who ρ_n and O_n are.
- Many titles in the bibliography are misspelled (capital vs. non-capital), such as those

containing Lieb-Robinson bounds.

Typos:

- Page 1, right column, paragraph 2, line 9, "mapping e.g. by trotterizing".
- Page 2, right column, paragraph 3, line 7, "n" hasn't been defined yet in the text.
- Page 2, right column, definition 1, ρ and ρ' are missing $_L$ in the equation.
- Page 5, right column, paragraph 6, line 8, "the errors in the energy density become independent".
- Page 6, left column, assumption 2, "satisfies the upper bound".
- Page 6, right column, paragraph 1, line 8, "Gaussian".
- Page 6, right column, section C, paragraph 1, line 1, "free-fermion" (sometimes in the text it's written as free fermion, and some others as free-fermion).
- Page 7, right column, paragraph 2, line 8, "larger than a constant".
- Page 8, left column, Eq. 11, $\rho_{\{L_n\}}$ instead of $\rho_{\{H_n\}}$.
- Page 8, right column, paragraph 3, there's a word missing after $\Theta(1/\epsilon^2)$.
- Page 8, right column, paragraph 4, line 1, "algorithms".
- Page 11, left column, paragraph 1, line 1, "results provide".
- Page 15, last paragraph. "corresponding".
- Page 16, proof, line 1, "starting point are".
- In general, many "." at the end of equations are missing throughout the appendix (such as in B4).
- Page 18, line 2, "rows".
- Page 20, line 3, "observable".
- Page 29, paragraph 3, line 5, $\text{norm}(L_{(j,\alpha)})$.
- In the next equation, who are the little k's? I guess they are the capital ones. Also in Eq. (2).
- Page 29, paragraph after Eq. (2), line 2, "Equation 2 is".
- Page 31, I guess ρ_0 and $\rho(0)$ are the same? Please unify.
- Sometimes you write Lieb-Robinson and some other Lieb Robinson, in case you want to unify it.
- Page 35, section D, line 2, "measured in the fixed point are robust".
- Page 36, "proof of proposition 8" (it's not clear, since the statement of the proposition is

so far away).

- Page 36, line 1, "is close to".
- Page 36, between (11c) and (12), $e_{\alpha,<}$ and $e_{\alpha,>}$.
- Page 36, (12), is it possible that it is a min instead of a max?
- Page 37, line 2, reference of the lemma missing.
- Page 39, paragraph 7, error E_n , and also in the equation after that.

Questions/suggestions:

- Is it possible to drop the assumption of unique fixed point for the rapidly mixing Lindbladians? Additionally, is it possible to weaken the assumption of rapid mixing to the mixing time provided only by a positive uniform gap in the Lindbladian, for example? I assume this would require proving Lieb-Robinson bounds for such Lindbladians.
- In general, I would suggest that you comment a bit on how restrictive are the assumptions you are imposing. As far as I have seen, there's only some comparison between how strong they are in the fermions and spins cases, but it is not explicitly mentioned why they are required for the proofs, or whether the authors believe that they are necessary at all.
- Proposition 7 can be improved by using tightening of the inequalities at several steps, such as Lemma 17 (where one can use quantum belief propagation to waive the dependence on the norm of H), and in the application of Lemma 18. In particular, since O has local support S_O , one can separate the interaction terms of H and H' intersecting S_O from those with empty intersection with it, in the proof of Proposition 7, to obtain bounds with exponential decay on the distance of the interaction terms to S_O for each of the terms in the latter set. This would allow the authors to get many decaying terms in the second summand of the second-to-last inequality of the proof, tightening the value of $f(\delta)$. Additionally, since all tools involved in the proof of this section have been recently extended to short-range interactions (i.e. exponentially decaying), they could enlarge the domain of applicability of their results.
- Is Proposition 11 very tailored to the embedded Glauber dynamics, or would something similar work as well for any other quantum rapidly mixing Lindbladian, possibly with some desirable properties?

Reviewer #3 (Remarks to the Author):

See the attached file.

[Editorial note: Please find it below.]

Nature Communications Review: Quantum advantage and stability to errors in analogue quantum simulators

Summary

First part of the paper: the authors are concerned with whether analogue quantum simulators can replicate the physics of local observables when errors are present in the simulator (errors here are defined relative to a target Hamiltonian, or introduced by Markovian or non-Markovian noise). The authors then go through multiple systems of interest and demonstrate stability of local observables under a variety of assumptions – most of which are standard/reasonable assumptions. A summary of these conditions has been given in Table 1. Notably the authors also consider robustness against non-Markovian errors.

Demonstrating stability is important as, without it, small perturbations of terms in the simulator (which may be due to noise) would invalidate conclusions we want to draw from analogue simulators. Thus, this is a well-motivated problem to study.

Second part of the paper: the authors argue that quantum simulators should be able to achieve a speedup relative to classical algorithms. However, this is not the traditional quantum advantage in terms of scaling the size of the system, but rather a quantum advantage in terms of the precision to which local observables are measured with respect to the value in the thermodynamic limit (the advantage of quantum simulators in terms of system-size is well known and heavily discussed). They argue this for the tasks of time-evolution simulation, ground state simulation, and simulation of fixed-points.

The motivation for quantum advantage in terms of $1/\epsilon$ scaling is not 100% clear to me – usually we are interested in a fixed precision, or $1/\text{poly}(n)$ precision. I feel that in practice the reason experimentalists take larger and larger system sizes is not to improve precision, but because at sufficiently large system size there may be qualitative changes in the behavior of the system (some sort of phase transition). In reality, they are likely to be happy with a fixed precision estimate of the error. Does anyone need precision beyond 5 significant figures for anything? Nonetheless, I think this is an interesting problem from a theoretical perspective, but maybe for a more niche audience who are more interested in complexity theory.

Broadly, the paper is well written and very understandable, and the authors should be congratulated on making this paper very readable.

Techniques

Stability of Local Observables (first part of the paper):

For the fermionic simulation, the authors demonstrate a set of new results for Gaussian fermionic Hamiltonians and their stability with respect to perturbations. I don't recognize the techniques and they appear to be both novel and non-trivial.

For the work on spin models, for ground state and thermal stability, the authors give an exposition of well known results about spectral flow and belief propagation developed by Hastings and utilized extensively by others. For the stability with respect to fixed points, the authors reuse techniques from Cubitt et al, but adapt it slightly to allow for non-Markovian error.

Quantum Advantage (second part of paper): the authors argue that there is an advantage to quantum simulators in terms of the scaling of the precision in the following way: the only classical algorithms which are demonstrated to converge are exact diagonalization or Krylov subspace methods, which one can then use to approximate the thermodynamic limit by taking larger and larger section of the lattice which we solve using these methods (this applies to time simulation as well as ground state simulation). To sustain their argument, they give an example of a Hamiltonian for which the necessary system size to reach error ϵ scales as roughly $n = \log(1/\epsilon)$.

For ground state simulation, the authors argue that one can beat this (assuming the expectation value converges exponentially fast to the value in the thermodynamic limit) using analogue simulation by using an adiabatic state preparation method, assuming a gap of $1/\text{poly}(n)$.

Comments on Stability of Local Observables Section

I have a problem with how some of the results in this work are presented: in particular, many of the results are stated in a way which suggests that they are new and have not been proven elsewhere in the literature. In particular, Proposition 6 is proven in the results they cite previously, but the authors state it as if it is new to their work and is being proven using from results by Hastings (e.g. saying that they use methods by Hastings, but not actually saying that this result has been proven before, e.g. in Appendix F.5 of arXiv:2106.12627, arXiv:quant-ph/0601019, etc.). *In my opinion, this is misleading.*

If this were a one off, I would forgive this and simply mention this in my minor comments section of this review, but the same problem occurs in Proposition 7: the stability of local observables for Gibbs states with exponentially decaying correlations has been proven in multiple previous works (arXiv:1309.0816, Sec. 2.3 arXiv:1609.07877, Lemma C.2 of arXiv:2301.12946, etc.) which the authors not only fail to cite, but seem to present this result as new. Furthermore, a result very similar to Proposition 2 has previously been proven by Hastings, but is not mentioned until near the end of the main text: "Stability of fermionic Hamiltonians with a spectral gap" [arXiv:1706.02270]. I am sure this is not intentional by the authors, and merely an artifact of how the results have been presented, nonetheless, I think this **must** be changed before publication anywhere.

Additionally, for Propositions 6 and 7, the authors appear to "reprove" these results in Appendix B and Appendix C, but also don't appear to add anything to the proofs which is not in previous works. If this is the case, then why are Appendices B and C here at all? In my opinion, at the very least

Propositions 6 and 7 should be written such that the references which they are from are stated in the “Proposition” environment as well as a reference in Table 1. Maybe they should even be removed and replaced with a reference to the relevant papers in Table 1 and a brief discussion in the text.

The results on stability of fixed points for spin models are “new” in sense that they allow for a non-Markovian term to be included which seems to be a straightforward (and small) change to the Cubitt et al. paper.

With this in mind, I really think that the authors should emphasize that the following results have already been proven and are not novel to their work:

1. *Stability of fermionic Hamiltonians with a spectral gap [arXiv:1706.02270].
2. **Stability of spin system time-dynamics with Markovian error [arXiv:1303.4744].
3. Stability of observables measured on ground states of gapped spin Hamiltonians [arXiv:1109.1588].
4. Stability of observables measured on Gibbs state of spin systems with exponentially decaying correlations [arXiv:1309.0816].
5. Stability of fixed points of spin systems that are rapidly mixing (but without non-Markovian effects) [arXiv:1303.4744].

Citations for all these works should also be included in Table 1.

*here the authors assume a different condition to this result by Hastings, but I still think the Hastings result should be clearly cited in the main text, and be in Table 1.

**to the authors’ credit, they reference this result well in the main text.

Comments on Quantum Advantage Section

The discussion in the quantum advantage section fairly “handwavy”. The authors are essentially arguing that because there are no currently known classical algorithms for computing these properties of analogue simulators, then there should be a quantum advantage. But they don’t add much new to the literature or provide much in the way of rigorous arguments for this. The idea that there is a quantum advantage for quantum simulators because we don’t have good classical algorithms so far is exactly the folklore knowledge that is typically used to justify building quantum simulators, and so I suspect the arguments are not novel here, or at least intuitively believed by the larger community. To be honest I find the arguments presented here fairly unconvincing (in that I agree with them, but they are fairly conventional and don’t appear to be particularly new).

I want to reiterate my skepticism towards the motivation here: I feel that in practice the reason experimentalists take larger and larger system sizes is not just precision, but because at sufficiently large system size there may be qualitative changes in the behavior of the system (e.g. as demonstrated in the work “Undecidability of the Spectral Gap” by Cubitt, Wolf and Perez-Garcia).

I think the authors should also probably add that we expect quantum advantage for quantum simulators in sense that preparing the ground state of a system with $1/\text{poly}(n)$ gap is expected to BQP-complete.

Conclusions

In my opinion this work adds the following new results to the literature:

1. Stability results for Gaussian fermionic Hamiltonians under new assumptions.
2. Stability with respect to non-Markovian errors.
3. A justification for studying quantum advantage with respect to the thermodynamic limit, and a semi-formalized discussion of this.

I think 1 is a genuinely new, interesting, and non-trivial set of results, but has been partial proven by Hastings before with different assumptions. 2 is new, but is a fairly straightforward adaptation of Cubitt et al.'s results. Point 3 is an interesting discussion, but I feel the arguments are not new or unique here – the justification that quantum simulators are good for simulating variables in the thermodynamic limit is well known and widely appreciated by the field. Indeed, the “Undecidability of the Spectral Gap” paper by Cubitt, Wolf and Perez-Garcia takes great pains to explain how their work prevents one from naïvely extending quantum simulation results by extrapolating to the thermodynamic limit. I agree with the authors’ point that this particular question is interesting from a complexity-theoretic perspective, but the authors themselves do little answer it. Perhaps the interest of point 3 to a broader audience and the author’s contributions are debatable here, and I leave it up to the editor to determine how interesting this section is.

While I do like the results 1 & 2, I am of the opinion that in their title and main text, the authors are overstating their own contributions to the problem of proving “Stability of Analogue Simulators to Errors”, much of which has been previously studied and I think these authors’ contributions are good, but are only a small part of the whole story, and the current article seems to obscure the contributions of other scientists. Furthermore, the main text is formulated in a way which sustains confusion about which contributions are novel. The discussion of quantum advantage for the thermodynamic limit is interesting, but I think perhaps for a more niche audience – it’s not clear that anyone is interested in computing the error of an observable to arbitrary precision.

Overall, I do not think the novel contributions of this paper are strong enough to be published in Nature Communications. If this paper is to be published – in Nature Communications or otherwise – I feel that the authors ought to change the title and edit the work to make it clearer what results are novel here, and which have been proven by previous authors.

Other Comments:

- “To compute the thermodynamic limit to a precision ϵ , we would first approximate the thermodynamic limit by a finite-size problem and then use these classical algorithms on the resulting finite-size problem.” To make this work, the authors have assumed that this is a valid thing to do (which is a very reasonable assumption!). It is probably worth mentioning that the

works arXiv:1502.04573, arXiv:1810.01858, arXiv:1910.01631 show that the problem of approximating a local observable to $O(1)$ precision is in fact an uncomputable problem in general.

- When talking about Exact Diagonalization and Krylov subspace methods, the authors should not a recent result arXiv:2311.18706 who present certified algorithms for observables of equilibrium systems in the thermodynamic limit. They do so in a different sense (they use a C^* -algebra formulation rather than taking the $N \rightarrow$ infinity limit) but I think it is still worth mentioning and citing.
- The authors are arguably not the first group to show stability with respect to critical systems. Coser & Perez-Garcia arXiv:1810.05092 show that systems which rapidly mix under a Lindbladian evolution (not from a trivial state) are stable with respect to perturbations in that direction. They demonstrate that this definition is equivalent to finite depth circuits in certain cases. Thus, given a critical system, there exists a set of states related by finite depth circuits which are stable with respect to perturbation. I believe this is also mentioned and used in arXiv:2311.07506 as part of their learning algorithm (I think they also correct a proof/assumption in the Coser & Perez-Garcia paper).
- Like the point above, in the Conclusions section, the authors write: “. Similarly, understanding the stability of Lindbladian dynamics and fixed point problems for quantum spin systems or non-Gaussian fermionic systems beyond the rapid mixing assumption would also be an important extension of our work.” – this problem is sort of studied in the works mentioned above arXiv:1810.05092, arXiv:2311.07506, and also recent work arXiv:2308.15495, all of which discuss this problem and its corresponding relationship with the definition of a phase of matter.

Minor Comments:

- In Definition 1, I'd appreciate it if the authors could include the definition of δ within the definition environment (currently it is only defined in text beforehand).
- The authors note that the stability of the spectral gap has only been proven for models with frustration free and LTQO. It might be worth adding it stability has been proven not to hold if these assumptions have not been met: arXiv:1502.04573, arXiv:1810.01858, arXiv:1910.01631.
- In Section V.A there is a missing reference in the first paragraph.

Reviewer 1

This is a report on the manuscript entitled "Quantum advantage and stability to errors in analogue quantum simulators". This work is addressing a timely question in quantum information science: It asks in what precise way we can hope analogue quantum simulators to have predictive power. At the heart of the matter is the issue of how errors in such prescriptions will build up, and what errors do to the quantum simulation.

After all, if they modify the results of the quantum simulation much, then there is little predictive power. Quantum error correction is out of scope for analogue quantum simulators, not only now, but presumably as a feature of such devices, as one deliberately tries to have high levels of control over the constituents, but does not aim at actually implementing quantum gates. Hence the stability to errors is a core research question in analogue quantum simulators.

The present work addresses this question. It is not the first manuscript discussing this question, but it adds a number of interesting technical points to it. On the highest level, the present work makes the point that errors tend to average away in a more beneficial fashion compared to quantum gate based quantum computing. The type of stability considered is a stringent one, namely a system-size independent notion of stability against extensive errors. This is a very stringent measure indeed, and it is not easy to prove such a stability for general quantum systems. For a very simple family of models, namely non-interacting or "Gaussian" settings, the stability in this sense is shown (and similar results are automatically inherited for some spin models).

This can be seen as an indication that this type of stability is realistic and is a solid start. What is interesting is that the criticality does not seem to matter much in the robustness, which may be seen as being surprising, in the light of the fact that correlations are long-ranged. It is further argued that this type of stability may lead to a quantum advantage, as it may indicate that this sort of stability is providing a kind of robustness that allows to make stable predictions, while classical simulation methods are more limited in time and system size.

I see the biggest strength of the manuscript being its conceptual clarity and the definition of the stringent robustness measure. The technical results reflect solid work, but may not be too surprising. The strongest limitation could well be the ansatz class considered, in that non-interacting or "free" or Gaussian models are considered for a substantial proportion of the manuscript, while for general systems, stronger assumptions are being made. Within the free fermionic framework, even open analogue quantum simulations can be considered, in which two types of noise compete with each other. It would have been better, possibly, to not say that this model is chosen "to be concrete", as it is said. But because the setting is simple in this way, and since one can make strong predictions, this setting may still provide a good indication of what the general setting would look like.

We thank the reviewer for their positive evaluation of our manuscript. Indeed, as the reviewer mentions, a large part of the manuscript deals with Gaussian models, which we expect to qualitatively capture the physics of more general models while at the same time being simple

enough to allow for tight results . We have followed the reviewer's suggestion and made a comment to this regard in the manuscript.

Definition 1 is the core definition of this work: It may be a good idea to state it in the trace distance, as this is basically what this definition is about. Maybe more seriously, it should be a bit more precisely stated, as the operator norm of O is this way not taken into account. There is still a kind of "strength" of the observable, which should be taken into account on the right hand side. This is an omission, also because earlier, the diamond norm (as a stabilized channel trace norm) is made use of, without mentioning and defining it.

We thank the reviewer for pointing out this omission. We have added explicitly the constraint that we assume $\|O\| \leq 1$ in definition 1. We have still retained the stability definition as a constraint on the perturbed observable, as opposed to (a trace norm) perturbation on the state since in almost all the problems that we consider, the state diverges much more strongly from the target state in the presence of errors, although certain (but not all) observables could still perturb modestly. We have also added a foot-note defining the diamond norm for readers not familiar with that notation.

Proposition 1 is interesting and insightful. And still, when it is said that the dynamics is stable to coherent and incoherent errors, it should be stated what kind of Gaussian errors this is stable against.

We thank the reviewer for this comment. We have emphasized, in the proposition statement, that the errors that we consider are of the form described in Eq. 4 and 5 in the text right before the proposition.

Assumption 1 is also very reasonable. While this is not proven for specific models, I agree that this is a reasonable assumption. It would be good to relate it to more commonly used assumptions, say, on energy spacings of non-integrable models. I also understand that this assumption includes gapped models. What is less clear is what kind of critical models are allowed for.

We believe that assumption 1 should be suggestive of models which have a gap between low energy eigenstates vanishing as $1/n$, where n is the system size. We have included this in the discussion around assumption 1. This is expected to be typically the case for many critical models in physics, such as those whose low energy physics is described by a conformal field theory - however, a more precise connection between this class of critical models to non-integrable models is an open question which we plan to address in future work.

Proposition 2 is then a bit unspecific: It would be good to be clear here also about the precise assumptions on the noise. Then, observables that are generated k -locally are introduced in passing: But they are very important for the type of robustness encountered, and maybe it would be good to spend a bit more time on their physical interpretation.

We thank the reviewer for this comment. The restriction on the observables is indeed very important and we have significantly expanded the discussion on the form of the observables considered in propositions 2-4, and moved it to the beginning of the subsection on equilibrium. We have given a physical interpretation of these observables as intensive observables which have contributions from all parts of the lattice (as opposed to being just localized at a few spatial locations). Furthermore, we

have also emphasized that if the equilibrium state (ground state, Gibbs state of fixed point) is itself translationally invariant, then the observables can always be taken to be translationally invariant, so this assumption is not necessarily restrictive for many problems of interest in physics.

Proposition 3 on Gibbs state preparation is equally interesting, but it would be good to specify the underlying assumptions a bit more precisely. In Proposition 3 it is clear that one cannot naively apply the results on ground states. But for gapped models with not too many spectral values above the gap one could presumably still draw this conclusion, so it may be nice to specify the connection between Proposition 2 and 3 a bit more.

We thank the reviewer for this comment. The only assumptions that we made in this proposition were on the observable and on the errors being coherent - we have clarified this in the proposition statement by referencing precise equations describing these assumptions. We have also expanded the discussion after the proposition to more precisely capture its relationship to the result on the ground state (proposition 2).

Proposition 4 seems incomplete: What does "in the fixed point of a spatially local free-fermion" mean?

We thank the reviewer for catching this typographical error. We have fixed it to "fixed point of a spatially local free-fermion Lindbladian" which should make the proposition statement unambiguous.

In Eq. (9), a different family of observables is considered. While this makes perfect sense from the perspective of the proof techniques explained in the appendix, it would be good to motivate this a bit more in the main text. The different dependence on the scaling in time and the dimension for spin models, can this be physically motivated?

We thank the reviewer for this comment - we have physically motivated the choice of observable (Eq. 11) from the fact that correlations spread from an observable to at most distance t during dynamics. We have also added a physical interpretation of the dependence on the t by interpreting it as the number of errors that occur in the volume of the light cone of a local observable.

To assume that the target Hamiltonian is stably gapped seems like a very strong assumption. This is indeed known for certain frustration free models with local topological order, but the perturbations allowed then are tiny. Would a relative perturbation also work?

We thank the reviewer for this comment. It is indeed true that the stability of the gap is very stringent, and is not rigorously known in many cases. Here, since our results are a direct consequence of the quasi-adiabatic continuation (or spectral flow method) previously developed by Hastings and Nachtergaele, we do not have a conclusive answer to how we can weaken this assumption. For the open system equilibrium problem (proposition 8), however, we can show stability without assuming robustness of the rapid mixing condition.

For Gibbs states having an exponential clustering of correlations (an s is missing here), this would basically mean that the temperature is high enough, right? The reference to 1-local observables is a

bit curious, however. I tried to derive this myself without looking at the appendix and did not see where the 1-locality assumption would come from. It might be good to explain better.

We thank the referee for noticing this subtlety - this arguably comes from the fact that the exponential clustering of correlations as an assumption is restricted to be between local observables, as opposed to observables that can have much larger non-local support. Consequently, we are restricted to only local observables. Since our previous notation of geometrically local observables as "1-local" could be misinterpreted as referring to observables with single site support, we have modified the statement of the proposition to make it clearer. For dynamics, as well as for the ground state problem (which is effectively mapped to a problem of dynamics via the spectral flow technique), a local observable when evolved in the Heisenberg picture is quasi-local. Consequently the stability result extends to k -local observables, which can be expressed in terms of a product of $O(1)$ local observables, since in the Heisenberg picture, each local observable in the product can be independently approximated by a quasi-local observable.

All this, of course, is the precursor of the discussion of presumably achieving a quantum advantage with analogue quantum simulators. This discussion starts out with discussing the ideal case - that of having no errors at all. This discussion of exact diagonalization techniques and Krylov subspace methods is insightful. The distinction between logarithmic convergence and power-law convergence is meaningful. What really matters, needless to say, is how noisy quantum simulators fare, and here the discussion becomes more vague. What follows is still a highly mature and insightful discussion: This is what we need to come closer to achieving realistic quantum advantages with analogue quantum simulators. The results are valid, original and of significance. There is a strong emphasis on problems that allow for a relatively simple solution, but this is due to the complexity of the problem. The approach taken is nevertheless expected to be helpful. I most enjoyed going through the appendices. While the results are not overly technically involved, I would think that the manuscript constitutes a solid discussion document. This is helped by the good table providing an overview. I hence tend to recommend publication.

Reviewer 2

The main aim of this paper is to study noisy quantum simulators to compute or approximate some relevant physical properties of many-body systems. These properties are studied for Gaussian fermions and for some spin models, and both in the cases of equilibrium and undergoing dynamics.

In the main text of the paper, the authors present a detailed discussion on the state of the art of analog quantum simulators, both motivating their use in various contexts, for different systems, as well as enlisting some of the main results for these objects the past few years. In particular, they discuss why a systematic and fundamental study of the quantum advantage for quantum simulators presents nowadays several theoretical issues, motivating the necessity to modify the standard definition in this framework. For that, they shift the study of many-body observables as a function of the system size to the intensive case, to end up showing that quantum simulators from some critical and non critical models are stable against errors.

First, they describe the notion of stability to consider in their quantum simulation tasks. Next, they address the case of Gaussian fermion models, for which they provide results both in the cases of finite-time dynamics and equilibrium. For the former, they show that measuring local Gaussian observables is stable to coherent and incoherent errors, whereas for the latter they study the analogous task measuring in the ground state and Gibbs state, respectively, showing stability with respect to coherent errors. This is subsequently extended to fixed points of rapidly mixing Lindbladians, allowing for coherent and Markovian incoherent errors.

In the next section, they present a collection of results for quantum spin systems, again both in the contexts of finite-time dynamics and equilibrium. The results shown are analogous to those for fermions, with the caveat that more restrictive assumptions have to be considered in the model. They conclude with a detailed section on quantum advantage with noisy quantum simulators, where they compare the ideal scenario with the noisy one.

Overall, I find the paper to be a very important contribution to the development of the field of quantum simulators, and I especially appreciate its careful writing. The main part of the manuscript is very enjoyable to read and the results are presented in a really clear way. The tables notably contribute to the understandability of the paper, and help the reader compare the results for both sets of models. For all this, I think that the paper should eventually be published in Nature Communications. However, before recommending acceptance, I have a few concerns about certain aspects of the proofs contained in the appendices that should be clarified by the authors.

Major concerns:

My main concern appear in Section III of the appendix, namely the “Stability of spin models”. In Lemma 12, they consider a family of operators $E'(s,t)$ and show an identity concerning them, as well

as the bosonic annihilation operators from the Lindbladian and the memory kernel. However, to prove it, they consider a superoperator which in particular contains $(E')^{-1}$. Why is this operator invertible? There are many examples in the literature in which this is a very subtle matter (e.g. in the context of adiabatic theorems for Lindbladians). Are the authors assuming anything in particular for L' so that the invertibility is guaranteed? This is not clear to me, and seems to be essential for the proof of this lemma and also for the rest of the section. One possibility is that they are only considering the “inverse” of the time-ordering, namely the reverse time ordering operator. In such a case, I don't see either how to obtain the derivative in the next line, since the A operators shouldn't commute with L' in general, but at least everything would be well defined. Could you please explain this point?

We thank the reviewer for this comment - indeed, as the reviewer points out, assuming that the channel $\mathcal{E}(s, t)$ is invertible is a non-trivial assumption and needs further justification. This is even more important for the problem at hand, where the Hilbert space is infinite-dimensional and operators appearing in the Lindbladian are unbounded. However the lemma used can be proved in a different way without resorting to inverting the channel $\mathcal{E}(s, t)$ but resorting to its trotterization - we have changed the proof of the lemma accordingly. We hope that this answers the referee's concern.

On Page 36, in the proof of the upper bound for $e_{\{\alpha, \succ\}}$, I find the whole argument quite confusing. First, after applying Lemma 19, I cannot see why you obtain Tr_E inside the norm of the second term. Secondly, the first term of that summand seems to disappear in the next line, and I cannot see why it should vanish without further explanation. Next, the last inequality just consists of adding positive terms that are also not justified. It would be great if you could give a further intuition/explanation of what is going on here.

We thank the reviewer for pointing out the issues with the presentation of these calculations. We have expanded the proof to include more steps and explanation of how we bound $e_{\{\alpha, \succ\}}$. We hope that the reviewer finds the new version of the proof more readable.

On Page 25, the equation after 8 is really confusing to me. First, if $t(x)$ is defined from the \tanh , with domain in \mathbb{R} , why do you need a 2π -periodic extension of it? Next, with this definition, I don't understand the last equality in the equation after 8. How can the expression with $t(x-y)$ and $t(x+y)$ coincide? Is it due to the product with the Dirichlet kernel? This is not immediately obvious to me from its expression.

We thank the reviewer for these comments. We use a 2π -periodic extension of $\tanh \beta x$ since we only need to use it for $|x| \leq \|\tilde{H}\| \leq O(1)$, and it is convenient to use a truncated Fourier series approximation to $\tanh \beta x$ to map the problem of observables in the Gibbs state to observables in dynamics (which we have already previously analyzed). We point out that it could be possible to rework the proof without resorting to a Fourier series representation and thus not requiring a periodic extension. Regarding the last equality in the equation after 8, that follows from a change of integration variable from y to $-y$ and using the fact that the Dirichlet kernel is an even function. We have added both of these remarks in the proof to make it easier to read and understand.

Minor concerns:

- In the notational preliminaries, you should mention the meaning of a norm without any subindex, since this is used quite frequently later.

We have added a note about the meaning of both vector and operator norms without a subindex in the notational preliminaries.

- On Page 2, definition 1, doesn't f need to be continuous or monotone, at least for n large enough?

We thank the reviewer for this comment. Indeed, f should be continuous at least around $\delta = 0$, else the definition would not be sensible. We had implicitly assumed it when requiring $f(\delta) \rightarrow 0$ as $\delta \rightarrow 0$ – we have now made this explicit in definition 1.

- On Page 20, the proof of Proposition 1 forces the reader to look for the definition of O in the main text. Maybe you could recall it here, so that it's easier to compare it to \tilde{O} .

We have repeated the definition of O in the proof to make it easier for the reader.

- On Page 20, the proof of Lemma 6 is slightly confusing with the change to the vectorized matrices and back. Maybe you could simplify it by directly working with the blocks and their norms. But the proof is perfectly fine like this, so of course feel free to leave it like this.

We thank the reviewer for this suggestion. We think that explicitly writing out the blocks could make it a bit more confusing and tedious, so we felt it is clearer to leave the proof as it is.

- On Page 20, Equation 2, should it be an inequality? Otherwise it's a really strong assumption.

We believe that the reviewer was referring to Page 29, Eq. 2 (there is no equation numbered 2 on Page 20). Indeed, as the reviewer correctly noted, this was a typographical error and it is an inequality and not an equality.

- Everywhere in Appendix III, and from hereafter, the numbering of the equations is different from that of the previous Appendix for fermions.

We thank the referee for catching this error - this was due to a typesetting error in LaTeX. We have fixed this issue and the numbering in the appendix should now be uniform.

- Page 39. The proof of proposition 9 is a bit confusing because of the order. It was mentioned that you would compute a lower bound for E_n two pages before, but it's only mentioned here towards the middle of the proof, and not in the statement of the proposition. If you recall it here, it's better for the understanding. It would also be better for the reader if you mentioned explicitly who ρ_n and O_n are.

We thank the reviewer for this remark. We have re-organized the appendix containing the proof of proposition 9 so as to move the model and its description right before the proposition statement and

proof. We have also made sure that all the quantities indexed by n (observables, states) are explicitly mentioned in the text. We hope that this makes the subsection clearer.

- Many titles in the bibliography are misspelled (capital vs. non-capital), such as those containing Lieb-Robinson bounds.

We thank the referee for this error - we have cross-checked all the references and made sure that they are consistent with the published papers.

Typos:

- Page 1, right column, paragraph 2, line 9, “mapping e.g. by trotterizing”.
- Page 2, right column, paragraph 3, line 7, “ n ” hasn’t been defined yet in the text.
- Page 2, right column, definition 1, ρ and ρ' are missing L in the equation.
- Page 5, right column, paragraph 6, line 8, “the errors in the energy density become independent”.
- Page 6, left column, assumption 2, “satisfies the upper bound”.
- Page 6, right column, paragraph 1, line 8, “Gaussian”.
- Page 6, right column, section C, paragraph 1, line 1, “free-fermion” (sometimes in the text it’s written as free fermion, and some others as free-fermion).
- Page 7, right column, paragraph 2, line 8, “larger than a constant”.
- Page 8, left column, Eq. 11, ρ_{L_n} instead of ρ_{H_n} .
- Page 8, right column, paragraph 3, there’s a word missing after $\Theta(1/\epsilon^2)$.
- Page 8, right column, paragraph 4, line 1, “algorithms”.
- Page 11, left column, paragraph 1, line 1, “results provide”.
- Page 15, last paragraph. “corresponding”.
- Page 16, proof, line 1, “starting point are”.
- In general, many “.” at the end of equations are missing throughout the appendix (such as in B4).
- Page 18, line 2, “rows”.
- Page 20, line 3, “observable”.
- Page 29, paragraph 3, line 5, $\text{norm}(L_{(j,\alpha)})$.
- In the next equation, who are the little k ’s? I guess they are the capital ones. Also in Eq. (2).
- Page 29, paragraph after Eq. (2), line 2, “Equation 2 is”.
- Page 31, I guess ρ_0 and $\rho(0)$ are the same? Please unify.
- Sometimes you write Lieb-Robinson and some other Lieb Robinson, in case you want to unify it.
- Page 35, section D, line 2, “measured in the fixed point are robust”.
- Page 36, “proof of proposition 8” (it’s not clear, since the statement of the proposition is so far away).
- Page 36, line 1, “is close to”.
- Page 36, between (11c) and (12), $e_{\alpha,<}$ and $e_{\alpha,>}$.
- Page 36, (12), is it possible that it is a min instead of a max?
- Page 37, line 2, reference of the lemma missing.
- Page 39, paragraph 7, error E_n , and also in the equation after that.

We thank the reviewer for their careful reading of our manuscript and for providing us with this detailed list of typographical errors. We have corrected these as well as any other errors we could find on a careful proofreading of the manuscript.

Questions/suggestions:

- Is it possible to drop the assumption of unique fixed point for the rapidly mixing Lindbladians? Additionally, is it possible to weaken the assumption of rapid mixing to the mixing time provided only by a positive uniform gap in the Lindbladian, for example? I assume this would require proving Lieb-Robinson bounds for such Lindbladians.

We thank the referee for this interesting question. One difficulty in dropping the assumption of a unique fixed point is that it then becomes hard to define a stability criterium on an equilibrium state. In particular, it could be that a Lindbladian does not have a unique fixed point, but under (sufficiently generic) local perturbations, its fixed point becomes unique. In this case, the long-time dynamics under the perturbed and unperturbed Lindbladian would be significantly different - for e.g. the unperturbed Lindbladian could even exhibit time-dependent evolution at longer times since it does not have a unique fixed point, while the perturbed Lindbladian would converge to its fixed point. However, if we make assumptions on the perturbations such that it does not change the dimensionality of the kernel of the Lindbladian (for e.g. those outlined in Simon Lieu et al, PRL (2020)), it could be possible to remove the assumption of a unique fixed point even while analyzing long-time dynamics.

Regarding the question of if only a spectral gap on the Lindbladian is enough, this also remains unclear. In particular, since the eigenvector matrix of a Lindbladian can be badly conditioned (for e.g. have a condition number that even scales exponentially with the system size), having a uniform spectral gap in the Lindbladian does not guarantee that that it mixes rapidly. There has been some progress in showing that a uniform spectral gap could be enough for Davies generators of commuting Hamiltonians (I. Bardet, PRL 2023), but the connection between spectral gaps and the mixing time of Lindbladians still remains to be clearly understood.

- In general, I would suggest that you comment a bit on how restrictive are the assumptions you are imposing. As far as I have seen, there's only some comparison between how strong they are in the fermions and spins cases, but it is not explicitly mentioned why they are required for the proofs, or whether the authors believe that they are necessary at all.

We thank the reviewer for this comment. While addressing their comment as well as the comments of the first reviewer, we have expanded the discussion of assumptions that we make in the manuscript.

- Proposition 7 can be improved by using tightening of the inequalities at several steps, such as Lemma 17 (where one can use quantum belief propagation to waive the dependence on the norm of H), and in the application of Lemma 18. In particular, since O has local support S_O , one can separate the interaction terms of H and H' intersecting S_O from those with empty intersection with it, in the proof of Proposition 7, to obtain bounds with exponential decay on the distance of the

interaction terms to S_{O} for each of the terms in the latter set. This would allow the authors to get many decaying terms in the second summand of the second-to-last inequality of the proof, tightening the value of $f(\delta)$. Additionally, since all tools involved in the proof of this section have been recently extended to short-range interactions (i.e. exponentially decaying), they could enlarge the domain of applicability of their results.

We thank the reviewer for this comment. We have followed the reviewer's instructions, as well as used better perturbation theory bounds for Gibbs states to significantly improve the stability estimate in proposition 7.

- Is Proposition 11 very tailored to the embedded Glauber dynamics, or would something similar work as well for any other quantum rapidly mixing Lindbladian, possibly with some desirable properties?

We thank the referee for this question. In proposition 11, we use the Glauber dynamics as a specific example of a rapidly mixing Lindbladian (since, even though it is a classical Markov process, it can always be embedded into a Lindbladian) which is also analytically solvable to allow us to provide lower bounds on the run-time of classical Krylov subspace methods. In principle, any other rapidly mixing Lindbladian should satisfy the same scaling as for this specific example, but providing a theoretically rigorous lower bound on the thermodynamic limit without an analytical closed form expression for the observable is difficult. Nevertheless, we believe that the question of finding "more quantum" Lindbladians which can be shown rigorously to rapidly mix is indeed an important question and is a problem that we (as well as other groups in the mathematical many-body community) are addressing as future work.

Reviewer 3

We thank the referee for their careful critique of our work. We believe that the referee has very precisely understood the technical content of our work, and has raised some important concerns regarding the presentation of the manuscript. We have edited our manuscript to incorporate the referee's comments and we believe that this has greatly improved the quality of the manuscript.

However, we respectfully disagree with the referee on their assessment of the novelty as well as the impact of our work, which we hope to clarify in this response. We will separately try to address the three points that the referee raised - on the novelty of our contribution, on the practical utility of studying quantum and classical run-times with precision and on our discussion of quantum advantage.

Regarding novelty of our contribution: First, we would like to address the referee's concern about the novelty of our technical contributions - the referee is of the opinion that while we make progress on stability results in many-body physics, our progress is incremental over previous work. The referee also feels that we have obfuscated previous contributions at several places in the manuscript. We would first like to clarify that that was never our intent, and we thought we had given credit to all the previous results on stability of many-body systems that we were aware of. We appreciate the referee's point that this was likely an unintentional error due to the presentation of the manuscript and trying to satisfy length constraints. Moreover, the referee has issues with only 2 (out of 8) stability results namely propositions 6 and 7, which study the impact of local perturbations in Hamiltonian (i.e. coherent errors) on the ground state and Gibbs state. In both of these results (and broadly in all the results for the spin systems), we had mentioned very explicitly in the manuscript that they follow straightforwardly from well known quasi-locality results in many body physics and did not intend to claim them as our own technical contributions. For e.g. in our view, proposition 6 (relating to ground states of gapped Hamiltonian) is simply a consequence of quasi-adiabatic continuation that has been extensively studied by Hastings and later by Nachtergaele and co-authors, and even in the very old seminal papers (Refs. [43, 46] of our paper), it was clear that the motivation was to make precise the notion that ground states of gapped Hamiltonians are stable phases of matter. Similarly, in the case of proposition 7, the stability result is a direct consequence of exponentially clustered correlations formalized and studied by Brandao (Ref. [40] of our paper), and we did nothing more than to use the lemma of this reference. Our intent behind including the appendices was to fill in the very minor details between these papers and our paper for the benefit of the reader, and we had not intended it to be a claim on these results as original technical contributions. Nevertheless, the referee did point out that these details have been filled in in papers on a different topic of quantum learning of many-body states. We have included them as references in our manuscript as

well and rewritten the section around proposition 6, 7 to reflect the fact that we are simply contextualizing and cross-examining these results for quantum simulators, and they are not novel from the perspective of mathematical many-body physics. Furthermore, as suggested by the referee, we have included some of the seminal references in the proposition statement and the summarizing table to emphasize even further that these results are not our main claim in the manuscript.

Even as per the referee, the remaining propositions are novel and in particular, the references on stability of Gaussian fermionic models are technically new contributions with entirely different proof techniques (Propositions 1 - 4). However, the referee still characterizes our work as being on a problem “much of which has been studied before”, and believes that we are overstating our contributions. We respectfully disagree with this characterization. In particular, for Gaussian models, the work by Hastings (Ref. [38] in our manuscript) proves the stability of the gap in a gapped model without the assumption of frustration freeness, which would then, using quasi-adiabatic continuation techniques, imply the stability of local observables. However, our proposition 2 is very different - first, it applies to gapless (thus critical) models, which is already a physically significantly different setting and its proof technique is entirely different from the methods used to prove the stability of gapped models. Importantly, in these models we do not have stability of local observables (as we discuss with the Anderson localization example), but we identify translationally invariant observables as a class where this does hold. This is an entirely new contribution in its own right and has a very strong experimental implication that goes beyond Hasting’s paper on Gaussian models - it suggests that, even in the presence of errors, quantum simulators could simulate many-body models that are *in a phase*, but also models that are *at a phase transition*. Furthermore, within the Gaussian case, we prove results (Propositions 1, 3, and 4) for dynamics, Gibbs states as well as open system fixed points (including for problems that are “critical” in a dissipative sense i.e. take polynomially long in system size to converge to the fixed point) which, to the best of our knowledge, have not been even partially considered before and are new both in terms of the physical implication of the result, as well as the proof technique used for these results.

Next, as the referee themselves point out, propositions 5 and 8 are novel in as far as they account for non-Markovian errors. We agree with the referee that the technical difficulty of these result is not as significant as our results on Gaussian fermion models (propositions 1 - 4), and from the fact that stability that was known to hold in the Markovian case (as shown in Ref. [37] of our manuscript which we had also explicitly cited), these are maybe even physically expected. We point out that arriving at this result does require an additional ingredient of input-output formalism to handle the possibly infinite-dimensional environment with unbounded interaction Hamiltonian on top of the tools used for the Markovian case, but we largely agree that the technical tools used in these two propositions are not remarkably new. However, we would emphasize that one of our goals with this paper is not just to add mathematical results on the question of stability of many-body models, but address the concrete and important experimental problem of quantum simulators. From the perspective of applicability of these results to experimentally realistic quantum simulators, we believe that extending stability results to the case of non-Markovian perturbations is a contribution that still addresses a physically important and experimentally relevant question.

Finally, the referee also comments that they find the title of the paper too broad, and in their opinion assumes unwarranted credit for the results in the paper. We hope to convince the referee that

this is not the case. In particular, we believe that in addition to adding many technically new results (Propositions 1 - 5, 8), our paper is the first paper that has contextualized stability results when applied to experimentally realistic analog quantum simulators. Our title does not claim to be the first paper to provide stability results for many-body physics, of which there are many older results. Furthermore, many of the references that the referee points us to for propositions 6, 7 are from a community (i.e. quantum learning) separate from the more experimentally inclined community of researchers in analog quantum simulators. We believe that our manuscript contextualizes these results for analog quantum simulation in particular, and bridges the gap between the community working on mathematical many-body physics and the community working on quantum implementations and analogue quantum simulation. From this perspective, we believe that our paper's title reflects our contribution, and does not subsume credit for past results (which would have been the case if, for instance, we had titled the paper as "Stability of many-body models in physics"). For this reason, we hope that the referee finds us retaining the title of the paper agreeable. Furthermore, for a similar reason, we have decided to retain the appendices with proofs of proposition 6, 7, not because we intend to claim them as novel contribution, but because we feel that a significant fraction of the readers interested in our paper would not be from the mathematical many-body physics or quantum learning community, but from the analog quantum simulation community who might appreciate a self contained document with all the important calculations in one manuscript. To this end, we point out that both referees 1 and 2 appreciated the presentation of our manuscript as a discussion document, and felt that it would be useful for the readers.

Assessing the quantum advantage by scaling with precision: The referee also raised a question on our basic premise - that for computational problems in many-body physics, we should study scaling of simulator (or classical algorithm) run-time with respect to the target precision and not with respect to the system size. They are of the opinion that, in practice, we only care about determining observables to some target precision (say to one digit of precision) and it is never really important to go to higher and higher precision, and consequently it is less important to study the scaling of the simulator run-time with the target precision. On the other hand, it is always important to go to larger and larger system sizes, which motivates the conventional framework of studying the run-time with system size.

However, we believe for many many-body physics problems (especially the classes of problems that we consider), these two notions are interlinked with each other, simply because interesting observables in most problems in many-body physics have a well-defined thermodynamic limit - consequently, continuing to increase the system size has increasingly less impact on the actual value of the observable. Consequently, from the referee's point of view, for these problems, even increasing the system size is irrelevant since after a certain system size, we more or less have a very good estimate of the observable. Any further increase in the system size would only make the computed observable come within a smaller target precision of the thermodynamic limit, but if only a certain number of digits of precision is demanded in the observable, studying the scaling with system size is also not practically interesting. In other words, for the problem class that we consider which also captures most practically interesting problems in many-body physics, studying scaling with precision and system size are related to each other. Furthermore, since we are interested in computing

the thermodynamic limit directly, then it would make sense to have a system-size agnostic notion of run-time scaling. As the referee is probably well aware of, this perspective has also been adopted in recent work on quantum Hamiltonian complexity [Refs. 35, 36].

In addition to our argument above, we point out that there are practically relevant scenarios where going to higher precision in the observable could be of interest. One such scenario would be computing critical values of the parameters at which phase transitions occur, which might require a relatively high precision in the observable being used to monitor the phase transition. Another example would be characterizing the critical exponents at a many-body phase transition, where we would likely need a high precision in the target observable in order to study the order of the phase transition.

On our contribution to the discussion on quantum advantage: Finally, the referee feels that our discussion in the section on quantum advantage is a relatively standard discussion and does not add anything new to the existing understanding of quantum advantage of quantum simulators. In particular, the referee feels that this section is qualitative (and that we do not make rigorous statements about quantum advantage), and is subsumed by previous similar discussion on this topic.

We thank the referee for this point. We agree with the referee that this section does not add any significantly new technical details to the manuscript - however, as also appreciated by referees 1 and 2, the idea behind this section was to lay out a discussion of the notion of quantum advantage in quantum simulators when accounting for (i) the fact that the run-time of quantum algorithms for problems in many-body physics that are concerned with calculating thermodynamic limits is more reasonably expressed in terms of precision and (ii) some of these simulation problems could be stable, and due to the reduced proliferation of errors, the thermodynamic limits could be well approximated on a noisy quantum simulator. We then intended to lay out estimates of run-times of classical algorithms for the problems that we had surveyed and proved stability results for, and argue that a noisy quantum simulator could provide an advantage as measured with respect to the hardware-limited precision in such problems. This section was intended to be a pedagogical discussion to lay out these ideas qualitatively and for a broader audience of experimental and theoretical physicists as well as for computer scientists working on both theoretical and empirical aspects of quantum simulation of many-body physics. We also point out that in a follow up work focussing on open system simulation, we have made many of the qualitative statements here precise and connected them to some standard complexity assumption [arXiv:2404.11081].

Technical and presentational comments

With this in mind, I really think that the authors should emphasize that the following results have already been proven and are not novel to their work:

1. Stability of fermionic Hamiltonians with a spectral gap [arXiv:1706.02270].
2. Stability of spin system time-dynamics with Markovian error [arXiv:1303.4744].
3. Stability of observables measured on ground states of gapped spin Hamiltonians [arXiv:1109.1588].
4. Stability of observables measured on Gibbs state of spin systems with exponentially decaying correlations [arXiv:1309.0816].
5. Stability of fixed points of spin systems that are rapidly mixing (but without non-Markovian

effects) [arXiv:1303.4744].

Citations for all these works should also be included in Table 1.

We have followed the reviewer's suggestion - in particular, proposition 6 and 7 now have one of the canonical references in the proposition statement, and we have also included a short statement in each row of Table 1 about how our work relates to previous work in different settings. We have also made more explicit in the text preceding propositions 6, 7 that the problem of stability of ground states and Gibbs states has been studied extensively in previous work, and that we only cross-examine these results from the perspective of analog quantum simulation. We hope that this satisfies the referee's concern about the presentation of our results.

“To compute the thermodynamic limit to a precision ϵ , we would first approximate the thermodynamic limit by a finite-size problem and then use these classical algorithms on the resulting finite-size problem.” To make this work, the authors have assumed that this is a valid thing to do (which is a very reasonable assumption!). It is probably worth mentioning that the works arXiv:1502.04573, arXiv:1810.01858, arXiv:1910.01631 show that the problem of approximating a local observable to $O(1)$ precision is in fact an uncomputable problem in general

We thank the reviewer for this suggestion, we have made a note of this subtlety in the manuscript.

When talking about Exact Diagonalization and Krylov subspace methods, the authors should not a recent result arXiv:2311.18706 who present certified algorithms for observables of equilibrium systems in the thermodynamic limit. They do so in a different sense (they use a C^* -algebra formulation rather than taking the $N \rightarrow \infty$ limit) but I think it is still worth mentioning and citing.

We thank the reviewer for this suggestion, we have cited this reference together with DS Wild et al (2022) on an algorithm for thermodynamic limit of observables in dynamics.

The authors are arguably not the first group to show stability with respect to critical systems. Coser & Perez-Garcia arXiv:1810.05092 show that systems which rapidly mix under a Lindbladian evolution (not from a trivial state) are stable with respect to perturbations in that direction. They demonstrate that this definition is equivalent to finite depth circuits in certain cases. Thus, given a critical system, there exists a set of states related by finite depth circuits which are stable with respect to perturbation. I believe this is also mentioned and used in arXiv:2311.07506 as part of their learning algorithm (I think they also correct a proof/assumption in the Coser & Perez-Garcia paper).

Like the point above, in the Conclusions section, the authors write: “. Similarly, understanding the stability of Lindbladian dynamics and fixed point problems for quantum spin systems or non-Gaussian fermionic systems beyond the rapid mixing assumption would also be an important extension of our work.” – this problem is sort of studied in the works mentioned above arXiv:1810.05092, arXiv:2311.07506, and also recent work arXiv:2308.15495, all of which discuss this problem and its corresponding relationship with the definition of a phase of matter.

We thank the reviewer for this suggestion, we have included these references in our outlook section.

Minor Comments:

In Definition 1, I'd appreciate it if the authors could include the definition of δ within the definition environment (currently it is only defined in text beforehand).

We thank the reviewer for this suggestion - we have mentioned in the definition that δ is the Hardware error rate in the simulator.

The authors note that the stability of the spectral gap has only been proven for models with frustration free and LTQO. It might be worth adding it stability has been proven not to hold if these assumptions have not been met: arXiv:1502.04573, arXiv:1810.01858, arXiv:1910.01631.

We thank the reviewer for this suggestion - we have incorporated these references into the main text.

In Section V.A there is a missing reference in the first paragraph.

We thank the reviewer for this suggestion. We have corrected this missing reference, as well as cross checked all the other references in the manuscript.

REVIEWERS' COMMENTS

Reviewer #1 (Remarks to the Author):

While my first report has been very long and detailed, this second one will be brief. I appreciate the careful changes made in the manuscript and the detailed responses to my report and those of the other reviewers. Some parts of the manuscript remain subtle, but I can see that it will eventually not be easily possible to weaken the assumptions made. For this reason, the settings considered are presumably good proxies for what is actually going on. I recommend publication of this good work.

Reviewer #2 (Remarks to the Author):

In their revised version, the authors have addressed all comments from my previous review and those from the other reviewers. Some of the new clarifications, explanations and improved estimates have, in my opinion, slightly increased the quality of the paper. I still believe this is a notable contribution that should be accepted for publication in Nature Physics.

I have mainly checked the parts in “red” and the comments from the authors in the response letter, but I’ve noticed that some modifications do not appear in red (e.g. in the statement of Proposition 7). Thus, it might be the case that I’ve missed some of these modifications. In any case, I have a few minor comments about the last additions to the paper:

- The proof of Lemma 12 is still very confusing in my opinion. On the first line, I assume you mean that you are defining D'_E , and not E' . And then, in the RHS of (C4), it should also be the same quantity as that defined on the previous line, so again $D'_E[\cdot, \cdot]$ and not $E'[\cdot, \cdot]$? Is the difference between both definitions that the authors want to evaluate $D'_E[\cdot, \cdot]$ in N and $E'(\cdot, \cdot)$ in R ? I am not entirely sure of how to check (C4), because I don't fully understand the quantities involved. Also, in (C5) there's also a “s” that is changed by “t” and the last line should have $C_{\{A_{j,\alpha}\}}$ instead of L . I would ask the authors to check the typos in this proof and clarify the notation (especially who D'_E is and how it related to E').

- Footnote 1 is slightly confusing, because of the change of typography between S and mathcal S. I assume it is to emphasise that the former corresponds to the latter otimes id without going into detail, but it might be better to just use the same letter for everything in a tiny abuse of notation.
- On page 5, after “Equilibrium”, there’s a “then” that should be “than”.
- Page 6, after Proposition 3, there’s a “Gibbs” misspelled.
- In Proposition 4, there’s a Lindbladian without capital.
- Page 26, there’s a Fourier without capital.

Concerning one of my questions from the previous review, the one regarding the possibility to weaken the assumption of rapid mixing to the mixing time provided only by a positive uniform gap in the Lindbladian, I appreciate the answer from the authors, but I wanted to further specify the reasoning lying behind my question. I didn’t mean that a uniform spectral gap in the Lindbladian could imply that it mixes rapidly, as that is a very hard problem and not true in general. However, in the reference cited by the authors (Cubitt et al, CMP 2015), when they consider the stability of the dynamics under Markovian perturbations in the Lindbladians, they assume that the original Lindbladian mixes rapidly for simplicity in the calculations, but comment that a polynomial mixing time (rather than polylog) might be enough to derive the similar results (see Section 4.5). Since it is known that a uniform spectral gap in the Lindbladian is sufficient to derive a polynomial mixing time (see e.g. (Kastoryano, Temme, JMP 2013, Theorem 22)), I was wondering whether assuming a uniform spectral gap in the original Lindbladian of your setting (rather than rapid mixing) would also yield stability results for the perturbations you consider in this paper. In any case, this is only a question/comment, and by no means I am asking the authors to include it in this paper.

Reviewer #3 (Remarks to the Author):

(see attached pdf)

[Editorial note: Please see below.]

I thank the authors for their changes and edits, particularly regarding the better referencing in new version of the work. In retrospect I think the paper does a good job the aim of addressing the concrete and important experimental problem of quantum simulators. With this in mind, I am happy with this paper being accepted to NComms.

Minor Comments:

- In Proposition 4, I think “lindbladian” should be capitalized.
- In their response, the authors write: “*In addition to our argument above, we point out that there are practically relevant scenarios where going to higher precision in the observable could be of interest*” and then list examples. It would be great to see these (or something like them) listed in the paper! Even copy-pasting what the authors wrote here would be good.
- [This is very much more of a suggestion rather than a requirement] With respect to the point above, if the authors choose to use the critical parameter estimation example, then maybe include some of the references: arXiv:2301.09369, arXiv:2202.02909, arXiv:2105.13350 just to show that others are expecting (a) quantum techniques to show some or of speed up & (b) the problem is (unsurprisingly) naturally a quantum problem. Unfortunately, I’m not aware of where the critical exponents problem has been studied from a quantum algorithms or complexity viewpoint. Perhaps this is evidence that the problem is best studied with quantum analogue simulators?

Reviewer 2

In their revised version, the authors have addressed all comments from my previous review and those from the other reviewers. Some of the new clarifications, explanations and improved estimates have, in my opinion, slightly increased the quality of the paper. I still believe this is a notable contribution that should be accepted for publication in Nature Physics.

I have mainly checked the parts in “red” and the comments from the authors in the response letter, but I’ve noticed that some modifications do not appear in red (e.g. in the statement of Proposition 7). Thus, it might be the case that I’ve missed some of these modifications. In any case, I have a few minor comments about the last additions to the paper.

We thank the reviewer for their careful reading of the revised manuscript. We had marked all the important edits in red – we believe that we had accidentally missed marking the changed estimate $f(\delta)$ in proposition 7, however this edited estimate was included in the marked table 1. So we believe that the reviewer has indeed carefully studied all the important changes in our manuscript.

The proof of Lemma 12 is still very confusing in my opinion. On the first line, I assume you mean that you are defining D'_E , and not E' . And then, in the RHS of (C4), it should also be the same quantity as that defined on the previous line, so again $D'_E[\cdot, \cdot]$ and not $E'[\cdot, \cdot]$? Is the difference between both definitions that the authors want to evaluate $D'_E[\cdot, \cdot]$ in N and $E'(\cdot, \cdot)$ in R ? I am not entirely sure of how to check (C4), because I don’t fully understand the quantities involved. Also, in (C5) there’s also a “s” that is changed by “t” and the last line should have $C_{\{A_j, \alpha\}}$ instead of L . I would ask the authors to check the typos in this proof and clarify the notation (especially who D'_E is and how it related to E').

We thank the reviewer for this comment and for their careful reading of the modified proof of Lemma 12. We have carefully checked the proof and corrected a small typographical error in C5 (now Eq. 20 of the supplement) which only slightly impacted our results by a multiplicative factor of $\frac{1}{2}$. $D'_E[\dots]$ is a discretized version of the channel $E'(\dots)$, and in the limit of the time-discretization going to 0, $D'_E[\dots] \rightarrow E'(\dots)$ as given in Eq. C4 (now Eq. 19 of the supplement). We hope that the revised proof is understandable to the reviewer.

Footnote 1 is slightly confusing, because of the change of typography between S and \mathcal{S} . I assume it is to emphasize that the former corresponds to the latter otimes id without going into detail, but it might be better to just use the same letter for everything in a tiny abuse of notation.

We thank the reviewer for pointing this out – we have edited it to uniformly use \mathcal{S} in the first footnote (which has been integrated into the main text as per the editorial instructions).

On page 5, after “Equilibrium”, there’s a “then” that should be “than”.

Page 6, after Proposition 3, there’s a “Gibbs” misspelled.

In Proposition 4, there’s a Lindbladian without capital.

Page 26, there’s a Fourier without capital.

We have fixed all the typographical errors that the reviewer pointed out.

Concerning one of my questions from the previous review, the one regarding the possibility to weaken the assumption of rapid mixing to the mixing time provided only by a positive uniform gap in the Lindbladian, I appreciate the answer from the authors, but I wanted to further specify the reasoning lying behind my question. I didn’t mean that a uniform spectral gap in the Lindbladian could imply that it mixes rapidly, as that is a very hard problem and not true in general. However, in the reference cited by the authors (Cubitt et al, CMP 2015), when they consider the stability of the dynamics under

Markovian perturbations in the Lindbladians, they assume that the original Lindbladian mixes rapidly for simplicity in the calculations, but comment that a polynomial mixing time (rather than polylog) might be enough to derive the similar results (see Section 4.5). Since it is known that a uniform spectral gap in the Lindbladian is sufficient to derive a polynomial mixing time (see e.g. (Kastoryano, Temme, JMP 2013, Theorem 22)), I was wondering whether assuming a uniform spectral gap in the original Lindbladian of your setting (rather than rapid mixing) would also yield stability results for the perturbations you consider in this paper. In any case, this is only a question/comment, and by no means I am asking the authors to include it in this paper.

We thank the reviewer for this insightful comment – the question that the reviewer asks is indeed very important and interesting. We hope to address it carefully in future work.

Reviewer 3

I thank the authors for their changes and edits, particularly regarding the better referencing in new version of the work. In retrospect I think the paper does a good job the aim of addressing the concrete and important experimental problem of quantum simulators. With this in mind, I am happy with this paper being accepted to NComms.

We thank the reviewer for their positive evaluation of our manuscript.

Minor Comments:

- In Proposition 4, I think “lindbladian” should be capitalized.

We have corrected the typographical error.

- In their response, the authors write: “In addition to our argument above, we point out that there are practically relevant scenarios where going to higher precision in the observable could be of interest” and then list examples. It would be great to see these (or something like them) listed in the paper! Even copy-pasting what the authors wrote here would be good.

•[This is very much more of a suggestion rather than a requirement] With respect to the point above, if the authors choose to use the critical parameter estimation example, then maybe include some of the references: arXiv:2301.09369, arXiv:2202.02909, arXiv:2105.13350 just to show that others are expecting (a) quantum techniques to show some or of speed up & (b) the problem is (unsurprisingly) naturally a quantum problem. Unfortunately, I’m not aware of where the critical exponents problem has been studied from a quantum algorithms or complexity viewpoint. Perhaps this is evidence that the problem is best studied with quantum analogue simulators?

We have followed the reviewer’s recommendation and added the following sentence (together with the suggested references) in the main text as a practical reason for trying to understand precision scaling of classical and quantum run-times.

Apart from being theoretically meaningful, expressing run-times in terms of precision might additionally be practically relevant in scenarios where we are trying to calculate either phase transition points \cite{bosse2024sketching, bausch2021uncomputability} or critical exponents characterizing a phase transition, both of which are typical calculations of interest in many-body physics.